# Relationship Prompt Learning is Enough for Open-Vocabulary Semantic Segmentation

**Jiahao Li**[1]**, Yang Lu**[1,2,3]**, Yuan Xie**[*4,5] **and Yanyun Qu**[*1,2,3]

[1]*School of Informatics, Xiamen University*
[2]*Institute of Artificial Intelligence, Xiamen University*
[3]*Key Laboratory of Multimedia Trusted Perception and Efficient Computing, Ministry of Education of China, Xiamen University*
[4]*School of Computer Science and Technology, East China Normal University*
[5]*Chongqing Institute of East China Normal University*

## Abstract

Open-vocabulary semantic segmentation (OVSS) aims to segment unseen classes without corresponding labels. Existing Vision-Language Model (VLM)-based methods leverage VLM's rich knowledge to enhance additional explicit segmentation-specific networks, yielding competitive results, but at the cost of extensive training cost. To reduce the cost, we attempt to enable VLM to directly produce the segmentation results without any segmentation-specific networks. Prompt learning offers a direct and parameter-efficient approach, yet it falls short in guiding VLM for pixel-level visual classification. Therefore, we propose the **R**elationship **P**rompt **M**odule (**RPM**), which generates the relationship prompt that directs VLM to extract pixel-level semantic embeddings suitable for OVSS. Moreover, RPM integrates with VLM to construct the **R**elationship **P**rompt **N**etwork (**RPN**), achieving OVSS without any segmentation-specific networks. RPN attains state-of-the-art performance with merely about **3M** trainable parameters (2% of total parameters).

## 1 Introduction

Open-vocabulary semantic segmentation (OVSS) [1–4] aims to segment novel classes without corresponding training images, which is still a challenging task in computer vision. Vision-Language Model (VLM) [5–7] has emerged as a powerful approach, acquiring comprehensive knowledge via large-scale image-caption matching training. Several VLM-based OVSS methods [8–10] achieve promising results. These methods employ the rich image-text representation knowledge inherent in VLM to improve segmentation performance and are categorized into two types: two-stage and one-stage methods. Two-stage methods [11–14] first generate image-level masks without semantics via a well-designed mask proposal network [15–18] and then classify these masks via the image-level classification ability of VLM. One-stage methods [19, 20] employ a semantic decoding network to distill VLM's comprehensive knowledge from the image to the pixel level, thereby producing pixel-level segmentation results.

However, both of these VLM-based methods rely on additional explicit segmentation-specific networks to obtain segmentation results, resulting in extensive training cost. To reduce the training cost, an intuitive idea is to make VLM directly produce segmentation results without these segmentation-

---

[*]Corresponding author: yxie@cs.ecnu.edu.cn, yyqu@xmu.edu.cn

38th Conference on Neural Information Processing Systems (NeurIPS 2024).

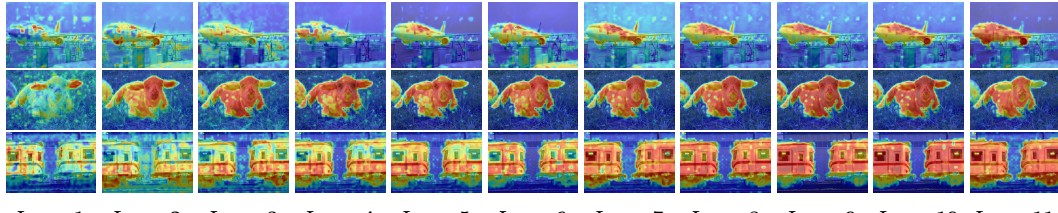

| Layer 1 | Layer 2 | Layer 3 | Layer 4 | Layer 5 | Layer 6 | Layer 7 | Layer 8 | Layer 9 | Layer 10 | Layer 11 |

Figure 1: **Visualization of relationship attention map** m **from the well-trained RPN.** The degree of attention from low to high is marked by colors from dark blue to red. The more attention, the darker red; the less attention, the darker blue. As the layers deepen, the attention maps exhibit increasingly precise pixel-level semantics. The images correspond to the attention maps for an aeroplane, a sheep, and a train, respectively.

specific networks[1]. In this context, prompt learning [5, 21–23] emerges as a practical approach, guiding VLM to transform image-text pair embeddings into pixel-level semantic embeddings suitable for OVSS, thereby directly achieving OVSS.

Without explicit segmentation-specific networks, applying prompt learning solely to VLM for OVSS is straightforward yet challenging. Typically, existing prompt learning methods fine-tune task-specific models to enhance performance on that task, yet they fail to secure cross-task performance gains. These methods employ either fixed or trainable vision prompt tokens [24], or they construct complex ViT-based networks for generating prompt [25, 26, 20, 27]. The limitations of such prompt include: 1) providing only image-level granularity, which restricts VLMs from performing tasks related to pixel-level visual classification, and 2) a lack of an image-text relationship, which hinders the exploration of VLMs' potential for open-vocabulary scene understanding. Therefore, it is difficult for these methods to enable VLM suitable for image-level classification to achieve open-vocabulary semantic segmentation directly.

In summary, addressing the above-mentioned issue lies in refining the granularity of prompt and strengthening the image-text relationship within them. By analyzing the outputs of VLM's encoder layer, we find they can construct an image-text relationship attention map via the attention mechanism, guiding the encoder to focus on relevant pixels. Thus, we propose the **R**elationship **P**rompt **M**odule (**RPM**) that utilizes the outputs of image and text encoding layers to enable pixel-level relationship prompting, enhancing prompt's granularity and image-text relationship. Moreover, we

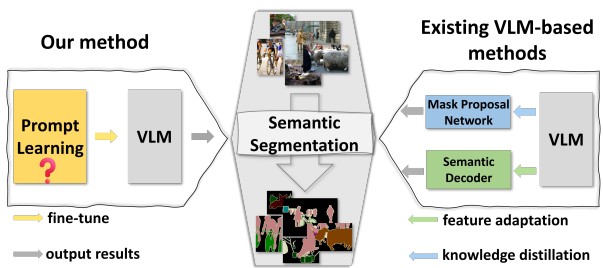

Figure 2: **Our method** *vs* **existing VLM-based methods.** Our method employs VLM to directly perform OVSS by prompt learning, while the other methods relies on additional explicit segmentation-specific networks.

implement a layer-by-layer guidance mode in VLM, enabling a progressive transfer of embeddings from the image to pixel level. As illustrated in Figure 1, each layer's relationship attention map is continuously refined following an image-to-pixel attention scheme. To obtain the segmentation results, we propose the **L**inear **P**rojection **M**odule (**LPM**) comprising merely two individual linear layers, which maps the image and text feature into a shared space, and then computes their Matrix product to produce the results. Finally, we propose the **R**elationship **P**rompt **N**etwork (**RPN**), which consists of RPM, LPM and VLM. Figure 2 shows the comparison between RPN and other OVSS methods. RPN employs VLM to directly output pixel-level predictions by prompt learning, while other methods use VLM to assist explicit segmentation-specific networks to obtain predictions. In these VLM-based methods, VLM transfers its rich knowledge to the mask proposal network by knowledge distillation or enables the semantic decoder to output segmentation masks by feature adaptation. The key difference between RPN and existing VLM-based methods is that RPN does

---

[1]To achieve the idea, some works train the VLM from scratch using pixel-level supervision, which relies on large-scale pixel-level pre-training. Note that we aim to employ a parameter-efficient fine-tuning method to reduce the need for additional segmentation-specific networks and large-scale pre-training.

not require any explicit segmentation-specific networks; it only adapts VLM to perform OVSS. Our contributions are summarized as follows:

- We propose the **R**elationship **P**rompt **M**odule (**RPM**), which generates pixel-level relationship prompt to guide VLM in transforming image-level embeddings to pixel-level ones suitable for OVSS.

- We propose the **R**elationship **P**rompt **N**etwork (**RPN**), employing prompt learning solely to adapt VLM for OVSS without explicit segmentation-specific networks.

- RPN attains state-of-the-art results on four public benchmarks by optimizing about **3M** trainable parameters (2% of total parameters).

## 2   Related Works

**Open-Vocabulary Semantic Segmentation.**   Open-vocabulary semantic segmentation [28, 29] aims to leverage the knowledge from representation distributions of seen categories to classify unseen categories. Existing methods can be divided into two types: generative and discriminative. Generative methods [30, 31, 28, 32] require the segmentation network to be aware of which categories are unseen during training, while discriminative methods [11, 8, 19, 33] directly transfer semantics from seen to unseen categories, which is more straightforward. SPNet [33] introduces the zero-shot task for the first time and proposes an end-to-end training paradigm, in which the visual embeddings are composed with the uniform semantic word embeddings to obtain the semantic logits. ZS3Net [31] utilizes a generative approach to project the text embeddings into visual space and generate visual embeddings for unseen categories. Subsequently, many works following the generative method have been proposed. STRICT [34] assumes that the pixels of unseen categories can be present during training the model and adopts the self-training strategy to optimize the model for classifying the unseen categories. GaCNet [30] proposes a novel context-aware feature generation method based on ZS3Net, in which pixel-wise contextual knowledge can be utilized to guide the feature generation process of unseen categories. CLIP-based approaches have also made great progress. ZegFormer [11] proposes two sub-tasks, i.e., class-agnostic grouping and segment-level zero-shot classification and presents the CLIP-based method for the first time. MaskCLIP [8] utilizes the frozen CLIP to make a minimal adaptation by fine-tuning a lightweight classifier and replacing it with that of the segmentation network. Zsseg [12] proposes a two-stage CLIP-based method, in which a proposal generator is used to generate binary masks and CLIP is required to classify them. ZegCLIP [19] presents a one-stage method in which CLIP directly transfers knowledge to a lightweight decoder.

**Vision-Language Models for Vision Tasks.**   Vision-language models for vision tasks [5, 21–23] are optimized with a large scale of image-text pair data on the internet. There are three categories: contrastive, generative, and aligned objectives. CLIP  [5] first proposes the paradigm of pre-trained vision-language model. DeCLIP [35] argues that CLIP is data-intensive and proposes a data-efficient training paradigm. UniCL [36] combines the two data sources to build a new image-text-label field and proposes unified contrastive learning. ZeroVL [21] proposes debiased sampling to deal with biased representation distributions and a new mixup method for the image and text models. OTTER [22] uses optimal transport to find the soft label for contrastive learning and handle the problem of noisy image-text pairs.

**Visual Prompt Learning.**   Visual Prompt Learning [37–39] is a technique that assists in adapting CLIP-like vision-language models for various visual tasks. CoOp [40] adopts trainable vectors as word prompt to adapt CLIP for vision classification. VP [41] utilizes perturbations as visual prompt. VPT [24] proposes trainable visual prompt to adapt each layer of the visual embeddings. UPT [38] constructs unified prompt modeling to extract trainable visual and textual prompt for adapting CLIP. MaPLe [39] adopts trainable prompt to guide both visual and textual embeddings and proposes a coupling function as a bridge to build a multi-modal prompt. DenseCLIP [26] uses the contextual information from the image to prompt the language model. Probabilistic prompt [20] applies multiple prompt sampled from probabilistic text embeddings to better understand the image. SegPrompt [27] proposes a category-level prompt to improve the model's class-agnostic segmentation ability.

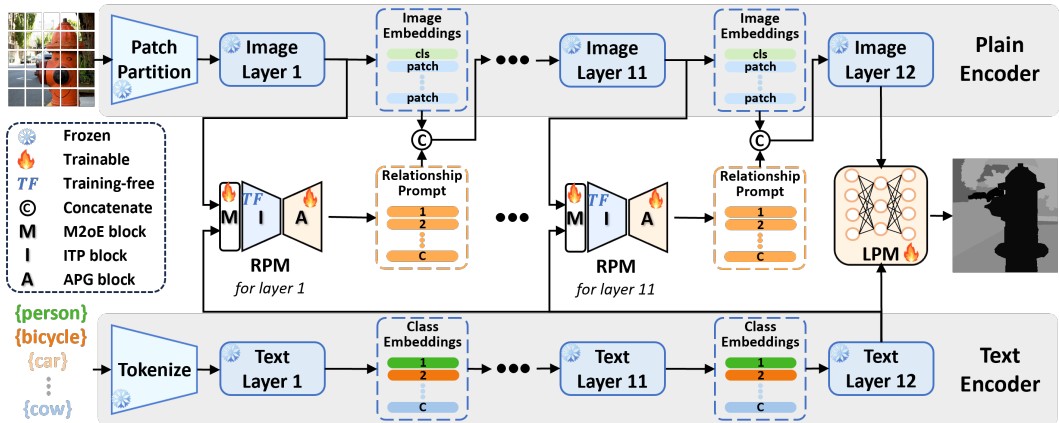

Figure 3: **Overview of RPN.** The end-to-end architecture is delineated into four principal components: 1) the frozen plain encoder, which adopts ViT architecture to encode visual knowledge with relationship prompt; 2) the frozen text encoder, which adopts CLIP text encoder architecture to encode class knowledge with text templates; 3) Relationship Prompt Module (RPM), which generates relationship prompt to guide the plain encoder to output pixel-level semantic embeddings; 4) Linear Projection Module (LPM), which consists of two individual linear layers to output OVSS results.

## 3 Approach

**Overview.** Our objective is to adopt prompt learning to develop a VLM-based OVSS method without any explicit segmentation-specific networks, thereby reducing training cost. As shown in Figure 3, the **R**elationship **P**rompt **N**etwork (**RPN**) is an end-to-end system comprising text, image and image-text relationship prompt branches. In the text branch, the text encoder inputs text to yield the class embeddings $\mathbf{t} \in \mathbb{R}^{C \times d}$, where $C$ and $d$ represent the number of classes and dimension, respectively. In the image branch, the plain encoder inputs images to obtain the image embeddings $\mathbf{v} \in \mathbb{R}^{(N+1) \times d}$, which includes the patch embeddings $\mathbf{p} \in \mathbb{R}^{N \times d}$ and the [CLS] token $\mathbf{g} \in \mathbb{R}^{1 \times d}$, with $N$ representing the number of patches. Concurrently, in the image-text relationship prompt branch, the proposed **R**elationship **P**rompt **M**odule (**RPM**) alongside the encoder takes the class and image embeddings to generate pixel-level relationship prompt, which is subsequently concatenated with the image embeddings to serve as the input for the next image layer. To obtain segmentation results, the last class and image embeddings are fed into the proposed **L**inear **P**rojection **M**odule (**LPM**) to calculate their Matrix product.

**Relationship Prompt Module.** RPM can guide the plain encoder of VLM to directly produce pixel-level semantic embeddings suitable for OVSS, due to its acquisition of three distinct types of knowledge. Firstly, it acquires multi-scale vision knowledge to locate objectives at distinct scales. Secondly, its image-text relationship knowledge enables the plain encoder to learn open-vocabulary semantics from text features. Thirdly, it introduces dynamic pixel-level knowledge, which is adaptive for the relationship knowledge, enabling pixel-level relationship prompt learning and thus transforming VLM's image-level embeddings into pixel-level ones. Therefore, RPM comprises three blocks, each dedicated to capturing one of the aforementioned knowledge types.

It is crucial for OVSS to obtain multi-scale image embeddings to locate targets of different scales. However, the image embeddings maintain a consistent scale across each image layer. Therefore, we propose multi-scale mixture-of-experts (M2oE) block to aggregate the patch embeddings across distinct scales. As illustrated in Figure 4, M2oE comprises a gating network and several

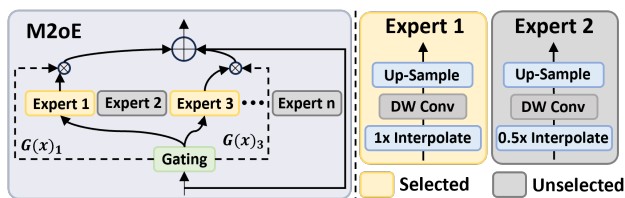

Figure 4: **M2oE.** $\otimes$ and $\oplus$ denote matrix product and addition.

expert networks as in [42, 43]. The gating network aims to dynamically activate different experts, each responsible for scaling the input to various extents. M2oE of the $i$-th layer is formulated as follows:

$$\mathbf{p}^i \leftarrow \mathbf{p}^i + \sum_{j=1}^{n} G(\mathbf{p}^i)_j \underset{\mathbf{r} \to \mathbf{d}}{\mathbf{Linear}}(E_j(\mathbf{p}^i)) \tag{1}$$

where $G(\cdot)$ and $E_j(\cdot)$ represent the gating network and the $j$-th expert of all $n$ experts, respectively. Note that the patch embeddings first reduce the dimension from $d$ to $r = 3$. And $\underset{\mathbf{r} \to \mathbf{d}}{\mathbf{Linear}}(\cdot)$ maps the output of each expert back to the original dimension $d$. See Appendix B for more details on M2oE.

It is crucial for OVSS to fuse the image embeddings with open-vocabulary semantics of the class embeddings. The key is to construct the image-text relationship that bridges the class and image embeddings. To achieve this, we propose image-to-pixel semantic attention (ITP) block, which utilizes the image embeddings and the last class embeddings to form the relationship attention map $\mathbf{m} \in \mathbb{R}^{N \times C}$. As illustrated in Figure 5, we first calculate the **Hadamard product** between the [CLS] token $\mathbf{g}$ and the last class embeddings $t$ after dimension alignment. Then, the **Matrix product** between the Hadamard product result and the patch embeddings $\mathbf{p}$ yields the relationship attention map $\mathbf{m}$. Consequently, the relationship attention map $\mathbf{m}^i$ of the $i$-th layer is formulated as follows:

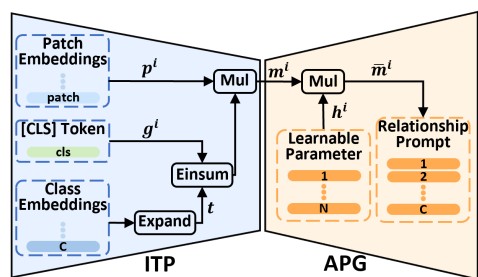

Figure 5: **ITP and APG.** Expand, Einsum and Mul denote expanding class dimension, Hadamard product and Matrix product.

$$\mathbf{m}^i = \mathbf{p}^i \cdot (t \odot \mathbf{g}^i)^\top \tag{2}$$

Intuitively, the first Hadamard product operation assigns weights to images in a batch, with their sum being one (the more important the image, the larger the weight), by fusing the class embeddings used to identify different classes and the [CLS] token used to identify each image in a batch, thus attaining image-level attention. The subsequent Matrix product operation weights the pixels, with the sum of pixel weights normalized to one (the more important the pixel, the larger the weight), by integrating the patch embeddings that contain pixel-level visual information, thus securing pixel-level attention. Therefore, we refer to the training-free operation (i.e., Eq. 2) as the image-to-pixel attention scheme, in which the first and the subsequent products extract image-level and pixel-level information, respectively. As illustrated in Figure 1, the relationship attention map construction process from the shallow to the deep layer demonstrates the effectiveness of the image-to-pixel attention scheme.

The construction of the adaptive image-text relationship for each pixel enables VLM to directly output pixel-level semantic embeddings. To achieve pixel-level dynamic tuning, we propose adaptive prompt generation (APG) block. As illustrated in Figure 5, we first initialize a trainable parameter $\mathbf{h} \in \mathbb{R}^{N \times d}$, representing the dynamic pixel-level knowledge. The adaptive relationship prompt $\bar{\mathbf{m}} \in \mathbb{R}^{C \times d}$ for each pixel is then derived from the Matrix product between the dynamic pixel-level knowledge $\mathbf{h}$ and the relationship attention map $\mathbf{m}$. Consequently, the adaptive relationship prompt $\bar{\mathbf{m}}^i$ of the $i$-th layer is formulated as follows:

$$\bar{\mathbf{m}}^i = \mathbf{m}^{i\top} \cdot \mathbf{h}^i \tag{3}$$

The role of the dynamic pixel-level knowledge $\mathbf{h}$ is twofold: 1) it projects the relationship attention map from a lower to a higher dimension to serve as an input for the plain encoder, 2) it fine-tunes the relationship prompt for each pixel. It is the fine-tuning of the relationship prompt for each pixel in a high-dimensional space that enables the plain encoder to directly obtain pixel-level semantics.

In addition, we integrate RPM in parallel within each layer of VLM. The image embeddings and the last class embeddings are fed into RPM to generate the relationship prompt, which is then merged with the image embeddings and fed into the next image layer. The prompt output from the image layer is discarded. The processing of each image layer is formulated as follows:

$$[\mathbf{v}^i, \_] = \mathbf{Layer}^i([\mathbf{v}^{i-1}, \bar{\mathbf{m}}^{i-1}]) \tag{4}$$

The notation $[,]$ represents the concatenation operation. See Appendix C for more details on RPM.

**Linear Projection Module.** Given that RPM enables the plain encoder to directly obtain pixel-level semantics, LPM aims to map the last image and class embeddings into a common space and calculate their Matrix product as the segmentation results. To this end, there are three intuitive designs as illustrated in Figure 6. All three designs share a common structure, consisting of an image branch and a text branch, each equipped with a linear layer and a normalization layer. The

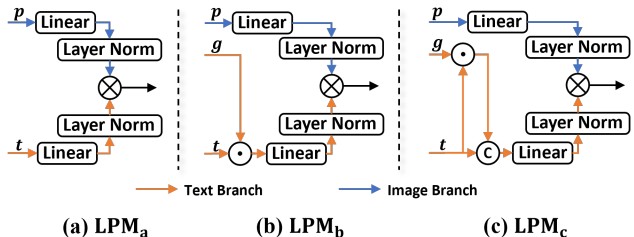

Figure 6: **Three kinds of LPMs.** $\odot$ and $\otimes$ denote element-wise product and matrix product.

image branch processing (the blue solid line) remains consistent, with the last patch embeddings $p$ sequentially passing through the linear layer and the normalization layer. The difference lies in the text branch processing (the orange solid line). In LPM$_a$ (as shown in Figure 6 (a)), only the last class embeddings $t$ are factored into the text branch. In LPM$_b$ (as shown in Figure 6 (b)), the last class embeddings $t$ and the last $\texttt{[CLS]}$ token $g$ first produce the Hadamard product before entering into the linear layer. In LPM$_c$ (as shown in Figure 6 (c)), the last class embeddings $t$ is concatenated with the Hadamard product result and then fed into the linear layer. Thus, the segmentation results $\mathbf{O}_a$, $\mathbf{O}_b$ and $\mathbf{O}_c$ of them are defined as follows:

$$\mathbf{O}_a = \mathbf{Linear}(p) \cdot \mathbf{Linear}(t)^\top \tag{5}$$

$$\mathbf{O}_b = \mathbf{Linear}(p) \cdot \mathbf{Linear}(t \odot g)^\top \tag{6}$$

$$\mathbf{O}_c = \mathbf{Linear}(p) \cdot \mathbf{Linear}([t \odot g, t])^\top \tag{7}$$

For simplicity, the normalization layers are omitted. We select LPM$_c$ as the proposed LPM experimentally (as shown in Table 6).

**Optimization.** There are two types of loss functions: the cross-entropy loss with the $\texttt{Softmax}$ function and the combination loss between the focal loss and the dice loss with the $\texttt{Sigmoid}$ function. The former employs one-hot encoding to render the class distribution as mutually exclusive in the embedding space, while the latter utilizes multi-label encoding to permit class distribution overlap. Practically, these losses are typically selected based on the used semantic decoders, such as the former with FPN[44] and the latter with a transformer decoder[45]. Considering that we employ LPM (comprising just two individual linear layers) to obtain the final segmentation results without any well-designed semantic decoders, we evaluate the aforementioned two loss functions. We refer to the former as the $\texttt{Softmax}$ loss and the latter as the $\texttt{Sigmoid}$ loss. We select the $\texttt{Sigmoid}$ loss experimentally (as shown in Table 6).

The $\texttt{Softmax}$ loss is formulated as follows:

$$\mathcal{L}_{softmax} = -\sum_{k=1}^{h \times w} y_k \cdot \log \hat{y}_k \tag{8}$$

where $y_k$ and $\hat{y}_k$ are the ground truth and the prediction vectors, respectively, and $h$ and $w$ represent the height and width of the input image. The $\texttt{Sigmoid}$ loss is formulated as follows:

$$\mathcal{L}_{focal} = -\sum_{k=1}^{h \times w} \alpha \cdot (\mathbf{1} - \hat{y}_k)^\gamma \cdot y_k \cdot \log(\hat{y}_k) \tag{9}$$

$$\mathcal{L}_{dice} = 1 - \frac{2 \sum_{k=1}^{h \times w} \hat{y}_k \cdot y_k^\top}{\sum_{k=1}^{h \times w} \hat{y}_k \cdot \hat{y}_k^\top + \sum_{k=1}^{h \times w} y_k \cdot y_k^\top} \tag{10}$$

$$\mathcal{L}_{sigmoid} = \lambda_1 \mathcal{L}_{focal} + \lambda_2 \mathcal{L}_{dice} \tag{11}$$

where $\alpha$, $\gamma$, $\lambda_1$ and $\lambda_2$ are hyperparameters. In Eq. 9, when $y_k$ equals the zero vector, $y_k$ and $\hat{y}_k$ are substituted with $\mathbf{1} - y_i$ and $\mathbf{1} - \hat{y}_i$, respectively.

# 4 Experiments

We evaluate our method in both zero-shot and open-vocabulary settings. See Appendix D for more details on the settings.

## 4.1 Implementation Details

**Datasets and Evaluation Metrics.** ADE20K[46] consists of 25k images for training and 2k images for validation. Pascal VOC 2012[47] includes 10,582 augmented training images and 1,449 validation images. COCO-Stuff164K[48] contains 118,287 training images and 5,000 validation images, with 171 classes in total. Pascal Context[49] consists of 10,100 images, of which 4,996 are used for training and 5,104 for validation, covering 60 classes. We employ pixel-wise classification accuracy (pAcc) and the mean of class-wise intersection over union (mIoU) for seen classes ($mIoU_s$), unseen classes ($mIoU_u$), and their harmonic mean (hIoU).

Table 1: **Efficiency comparison with state-of-the-art methods. #Params(M)** represents the total number of trainable parameters.

| Methods | #Params(M) | FLOPs(G) | FPS |
|---|---|---|---|
| **VOC** | | | |
| ZegFormer [11] | 60.3 | 1829.3 | 1.7 |
| ZegCLIP [19] | 13.8 | 110.4 | 9.0 |
| **RPN(ours)** | **3.2** | **84.2** | **10.6** |
| **COCO** | | | |
| ZegFormer [11] | 60.3 | 1875.1 | 1.5 |
| ZegCLIP [19] | 14.6 | 123.9 | 6.7 |
| **RPN(ours)** | **3.2** | **101.3** | **10.2** |

**Training Strategy.** We conduct all experiments on eight NVIDIA GTX 3090 GPUs using the MMSegmentation tool [50]. If not specified, we employ the pre-trained CLIP ViT-B/16 model for both the plain encoder and the text encoder. We set the batch size of 4 for each GPU and set the input resolution to $512 \times 512$ pixels. The data augmentation strategy adheres to the default settings in MMSegmentation, which includes random image resizing with a short-side range of [256, 1024] and a crop size of $512 \times 512$. The optimizer is AdamW, initialized with a learning rate of $2 \times 10^{-5}$ and a weight decay of $1 \times 10^{-2}$. The learning rate follows a polynomial decay schedule with a power of 0.9. The number of iterations is set to 20K for the VOC dataset, 80K for the COCO dataset, and 40K for the Context dataset. We set $\lambda_1$ and $\lambda_2$ in Eq. 11 to 100 and 1, respectively.

## 4.2 System Level Comparison

**Efficiency Comparison.** We present an efficiency comparison with state-of-the-art methods in Table 1. The results of compared methods are derived from [19]. To ensure a fair comparison, we report our results based on the open-source code from [19] and evaluate them with an input resolution of $512 \times 512$ on a single NVIDIA GTX 1080 Ti GPU. Our method outperforms the other methods in efficiency, achieving the lowest number of trainable parameters and the smallest FLOPs.

**Comparison in the Zero-Shot Setting.** We show the performance comparison with the state-of-the-art methods in the zero-shot setting in Table 2, and conduct the comparison under three scenarios: without self-training, with self-training, and fully supervised. In the absence of self-training, our method surpasses FreeSeg [10] with +6.0% $mIoU_u$ on the VOC dataset and +0.6% $mIoU_u$ on the COCO dataset. With self-training, our method outperforms ZegCLIP [19] with +3.7% $mIoU_u$ on the VOC dataset, +1.3% $mIoU_u$ on the COCO dataset, and +2.3% $mIoU_u$ on the Context dataset. Under the fully supervised scenario, our method exceeds ZegCLIP [19] with an average of +2.0% $mIoU_u$ across the three datasets. We attribute the modest improvement on the COCO dataset to the bias from the extremely unbalanced training and validation set ratios (more training data and less validation data), which contrasts with the performance on the Context dataset (less training data and more validation data), reflecting the robust zero-shot learning ability of our method.

**Comparison in the Open-Vocabulary Setting.** We show the performance comparison with the state-of-the-art methods in the open-vocabulary setting in Table 3. Our method does not require an additional training dataset. The results indicate that no method can consistently outperform others across all validation datasets; however, our method attains state-of-the-art performance on the A-847, A-150 and the PAS-20 datasets. As analyzed in [53], the Context dataset and the ADE20K dataset

Table 2: **Performance comparison in the zero-shot setting (unit:%).** Here, the best results are shown in bold and the second-best results are underlined. The self-training represents applying self-training via generating pseudo labels on all unlabeled pixels like in [19, 8]. The symbol '†' indicates pseudo labels are merely annotated on unseen classes pixels excluding the ignore part.

| Methods | VOC | | | | COCO | | | | Context | | | |
|---|---|---|---|---|---|---|---|---|---|---|---|---|
| | pAcc | mIoU$_s$ | mIoU$_u$ | hIoU | pAcc | mIoU$_s$ | mIoU$_u$ | hIoU | pAcc | mIoU$_s$ | mIoU$_u$ | hIoU |
| *w/o self-training* | | | | | | | | | | | | |
| ZegFormer [11] | - | 86.4 | 63.6 | 73.3 | - | 36.6 | 33.2 | 34.8 | - | - | - | - |
| ZegFormer+MAFT [51] | - | 91.5 | 80.7 | 85.7 | - | 36.4 | 40.1 | 38.1 | - | - | - | - |
| ZSSeg [12] | 90.0 | 83.5 | 72.5 | 77.5 | 60.3 | 39.3 | 36.3 | 37.8 | - | - | - | - |
| ZSSeg +MAFT [51] | - | 87.1 | 76.1 | 81.2 | - | 36.1 | 35.9 | 36.0 | - | - | - | - |
| ZegCLIP [19] | 94.6 | 91.9 | 77.8 | 84.3 | 62.0 | 40.2 | 41.4 | 40.8 | 76.2 | 46.0 | 54.6 | 49.9 |
| FreeSeg [10] | - | 91.9 | 78.6 | 84.7 | - | **42.4** | 42.2 | **42.3** | - | - | - | - |
| **RPN(ours)** | **95.8** | **93.1** | **84.6** | **88.6** | **64.4** | 40.8 | **42.8** | 41.8 | **76.4** | **47.7** | **58.7** | **52.6** |
| *w/ self-training* | | | | | | | | | | | | |
| ZegCLIP [19] | 95.1 | 91.8 | 82.2 | 86.7 | 68.8 | 40.6 | 54.8 | 46.6 | 77.2 | 46.6 | 65.4 | 54.4 |
| MaskCLIP† [8] | - | 88.8 | 86.1 | 87.4 | - | 38.1 | 54.7 | 45.0 | - | 44.4 | 66.7 | 53.3 |
| ZegCLIP† [19] | 96.2 | 92.3 | 89.9 | 91.1 | 69.2 | **40.7** | 59.9 | 48.5 | 77.3 | 46.8 | 68.5 | 55.6 |
| **RPN†(ours)** | **97.1** | **93.1** | **93.6** | **93.3** | **69.3** | 40.6 | **61.2** | **48.8** | **78.3** | **48.1** | **70.8** | **57.3** |
| *fully supervised* | | | | | | | | | | | | |
| ZegCLIP [19] | 96.3 | 92.4 | 90.9 | 91.6 | 69.9 | 40.7 | 63.2 | 49.6 | 77.5 | 46.5 | 78.7 | 56.9 |
| **RPN(ours)** | **97.2** | **94.0** | **94.6** | **94.3** | **70.8** | **41.1** | **64.1** | **50.5** | **78.7** | **48.5** | **80.1** | **60.4** |

Table 3: **Performance comparison in the open-vocabulary setting (unit:%).** Here, the best results are shown in bold and the second-best results are underlined.

| Methods | VLM | Training Set | A-847 | PC-459 | A-150 | PC-59 | PAS-20 |
|---|---|---|---|---|---|---|---|
| OVSeg [14] | ViT-B/16 | COCO-Stuff+COCO Caption | 7.1 | 11.0 | 24.8 | 53.3 | 92.6 |
| CAT-Seg[52] | ViT-B/16 | COCO-Stuff | 8.4 | 16.6 | 27.2 | **57.5** | 93.7 |
| SAN[53] | ViT-B/16 | COCO-Stuff | 10.1 | 12.6 | 27.5 | 53.8 | 94.0 |
| SED[54] | ConvNeXt-B | COCO-Stuff | 11.4 | **18.6** | **31.6** | 57.3 | 94.4 |
| **RPN(ours)** | ViT-B/16 | COCO-Stuff | 11.4 | 17.3 | 31.5 | 57.1 | **95.2** |
| OVSeg [14] | ViT-L/14 | COCO-Stuff+COCO Caption | 9.0 | 12.4 | 29.6 | 55.7 | 94.5 |
| CAT-Seg[52] | ViT-L/14 | COCO-Stuff | 10.8 | 20.4 | 31.5 | **62.0** | 96.6 |
| SAN[53] | ViT-L/14 | COCO-Stuff | 13.7 | 17.1 | 33.3 | 60.2 | 95.5 |
| FC-CLIP[55] | ConvNeXt-L | COCO Panoptic | 14.8 | 18.2 | 34.1 | 58.4 | 95.4 |
| SED[54] | ConvNeXt-L | COCO-Stuff | 13.9 | **22.6** | 35.2 | 60.6 | 96.1 |
| **RPN(ours)** | ViT-L/14 | COCO-Stuff | **14.9** | 22.1 | **36.4** | 61.9 | 96.6 |

exhibit the highest and lowest label-set similarities with the training dataset, respectively. Therefore, our method showcases a more comprehensive open-vocabulary learning ability.

## 4.3 Ablation Study

We conduct the ablation experiments on the VOC and the COCO datasets. If not specified, most are conducted on the VOC dataset. See Appendix D for more details on the experiments.

**Image-to-Pixel Attention Scheme.** RPM aims to transform image-level embeddings from VLM into pixel-level semantic embeddings, enabling direct OVSS. To illuminate its functionality, we employ the Mean Attention Distance (MAD) [56, 57] as a metric, reflecting the granularity of information aggregated within the self-attention head. As illustrated in Figure 7, we analyze MAD of each self-attention head during the initial and advanced stages of training. A higher point indicates a larger receptive field, and greater point spacing signifies richer feature diversity. In the initial training phase, shallow and deep layer information exhibit marked differences: the former concentrates on the local field with fine granularity and high diversity, while the latter focuses on the global field with coarse granularity and limited diversity. During training with RPM, deep layer information maintains attention on both local and global fields without sacrificing granularity or diversity. Clearly, deep layer information, augmented with relationship prompt learning, is more suitable for pixel-level semantic segmentation tasks, thereby diminishing the need for additional segmentation-specific networks. In addition, we visualize the relationship attention map in Figure 1. The first line denote the relationship attention maps for seen classes across each layer; the last two lines for unseen classes. For seen classes, the model has prior pixel-level semantic, so the relationship attention map only needs to focus on a few pixels to guide the model to make predictions for these pixels (e.g., the relationship attention map for airplane has fewer highlighted areas). For unseen classes, the model lacks corresponding semantic, so the relationship attention map needs to focus on more pixels to provide the model with

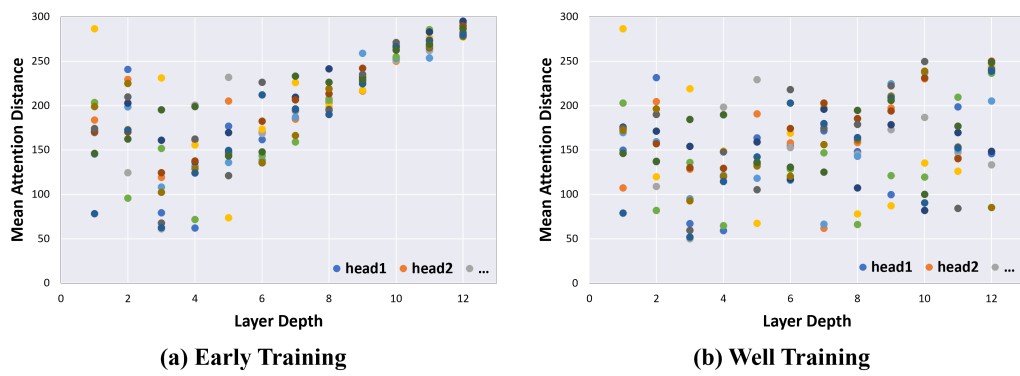

|     (a) Early Training     |     (b) Well Training     |

Figure 7: **Mean Attention Distance of each self-attention head.**

Table 4: **Impact of different modules (unit:%).** Methods without LPM represent eliminating the linear layers in LPM, i.e., discarding **Linear**$(\cdot)$ in Eq.7.

| RPM | | LPM | VOC | | | | COCO | | | |
|---|---|---|---|---|---|---|---|---|---|---|
| w/ M2oE | w/o M2oE | | pAcc | mIoU$_s$ | mIoU$_u$ | hIoU | pAcc | mIoU$_s$ | mIoU$_u$ | hIoU |
| | | | 77.1 | 76.3 | 14.3 | 24.1 | 48.3 | 31.8 | 16.4 | 21.6 |
| | ✓ | | 94.7 | 92.1 | 81.2 | 86.3 | 63.3 | 39.2 | 40.3 | 39.7 |
| ✓ | | | 95.1 | 92.4 | 81.7 | 86.7 | 63.7 | 39.5 | 40.3 | 39.9 |
| | | ✓ | 88.8 | 87.3 | 45.3 | 59.6 | 49.3 | 31.3 | 20.3 | 24.6 |
| | ✓ | ✓ | 95.6 | 92.9 | 83.4 | 87.9 | 64.1 | 39.7 | 41.6 | 40.6 |
| ✓ | | ✓ | 95.8 | 93.1 | 84.6 | 88.6 | 64.4 | 40.8 | 42.8 | 41.8 |

more sufficient pixel-level semantic (e.g., the relationship attention map for sheep highlights the complete semantic at shallow layers).

**Impact of Different Modules.** Table 4 shows the impact of various modules. RPM (the first line) denotes the combination of M2oE, ITP and APG. RPM without M2oE (the second line) denotes the ablation about APG and ITP. Utilizing RPM and LPM yields the best performance. The performance improvement attributed to RPM (+62.6%) is significantly greater than that of LPM (+35.5%). Furthermore, integrating LPM on top of RPM yields a modest performance gain of 1.9%. In contrast, incorporating RPM based on LPM results in a substantial 29% improvement. Therefore, we conclude that VLM with relationship prompt learning is enough for OVSS without any explicit segmentation-specific networks.

**Exploration of Different Designs in RPM.** Firstly, we explore various designs of the ITP block. As discussed in Section 3, removing the patch embeddings, which contain pixel-level visual information, results in a loss of pixel-level attention for ITP. Conversely, omitting the [CLS] token, responsible for identifying each image, results in a loss of image-level attention. To assess the significance of the image-to-pixel attention scheme, we define the 'without pixel-level attention' and the 'without image-level attention' scenarios by excluding the patch embeddings $\mathbf{p}^i$ and the [CLS] token $\mathbf{g}^i$ from Eq. 2, respectively. Note that employing the 'without pixel-level attention' scenario necessitates altering the dimension of

Table 5: **Ablation study on different designs in RPM (unit:%).**

| | Attention | pAcc | mIoU$_s$ | mIoU$_u$ | hIoU |
|---|---|---|---|---|---|
| ITP | w/o image-level | 73.1 | 74.2 | 21.4 | 33.2 |
| | w/o pixel-level | 87.7 | 77.6 | 68.1 | 72.5 |
| | w/ both | 95.8 | 93.1 | 84.6 | 88.6 |
| | **Modes** | **pAcc** | **mIoU$_s$** | **mIoU$_u$** | **hIoU** |
| APG | nn.Linear | 95.1 | 92.1 | 80.6 | 86.0 |
| | nn.Parameter | 95.8 | 93.1 | 84.6 | 88.6 |
| | **Dimension** | **pAcc** | **mIoU$_s$** | **mIoU$_u$** | **hIoU** |
| M2oE | $r=3$ | 95.8 | 93.1 | 84.6 | 88.6 |
| | $r=6$ | 95.8 | 93.4 | 84.5 | 88.7 |
| | $r=12$ | 95.9 | 93.8 | 84.3 | 88.8 |
| | **Modes** | **pAcc** | **mIoU$_s$** | **mIoU$_u$** | **hIoU** |
| | Multi-Scale | 94.9 | 92.8 | 84.1 | 88.2 |
| | M2oE | 95.8 | 93.1 | 84.6 | 88.6 |

the trainable parameter $\mathbf{h}^i$ from $\mathbb{R}^{N \times d}$ to $\mathbb{R}^{d \times d}$. The results presented in Table 5 (ITP) corroborate the efficacy of the image-to-pixel attention scheme. Secondly, we explore different configurations of the APG block and evaluate two distinct trainable modes: nn.Linear and nn.Parameter. The outcomes in Table 5 (APG) advocate for the implementation of the nn.Parameter mode. Thirdly, we explore the dimension $r$ of M2oE and different multi-scale aggregation modes in Table 5 (M2oE). The results show that performance enhancements are marginal with increasing dimensions. Given the trade-off between performance gain and parameter increase, we adopt $r = 3$. The 'Multi-Scale' mode refers to removing the gating network and directly aggregating the features processed by all expert networks (See Appendix B for more details).

**Exploration of Different Designs in LPM.** We evaluate three different LPM designs (shown in Figure 6) using `Softmax` and `Sigmoid` losses. The results in Table 6 reveal that $\text{LPM}_c$ with `Sigmoid` loss is the most effective strategy. In addition, using `Sigmoid` loss is significantly better than using `Softmax` loss. This reflects that the relationship prompt does not directly focus on pixels, but follows the image-to-pixel process.

Table 6: **Ablation study on three different LPMs using** `Softmax` **and** `Sigmoid` **losses (unit: %).**

| $\text{LPM}_a$ | $\text{LPM}_b$ | $\text{LPM}_c$ | pAcc | $\text{mIoU}_s$ | $\text{mIoU}_u$ | hIoU |
|---|---|---|---|---|---|---|
| \multicolumn{7}{c}{Using `Softmax` loss} |
| ✓ | | | 70.6 | 67.3 | 19.6 | 30.3 |
| | ✓ | | 89.4 | 88.6 | 55.5 | 68.2 |
| | | ✓ | 90.4 | 89.5 | 59.3 | 71.4 |
| \multicolumn{7}{c}{Using `Sigmoid` loss} |
| ✓ | | | 87.5 | 84.6 | 61.4 | 71.2 |
| | ✓ | | 95.3 | 93.1 | 83.6 | 88.1 |
| | | ✓ | 95.8 | 93.1 | 84.6 | 88.6 |

**Impact of Training-Free Projection Modules.** Note that our method comprises two trainable modules: RPM and LPM. To further explore the performance of RPM, we eliminate all linear layers in LPM to construct three variants of training-free projection networks. Following the sequence depicted in Figure 6, we denote the training-free projection modules as $\text{TFPM}_a$, $\text{TFPM}_b$ and $\text{TFPM}_c$. The results in Table 7 demonstrate that both $\text{TFPM}_b$ and $\text{TFPM}_c$ can achieve the state-of-the-art performance. The suboptimal results with $\text{TFPM}_a$ suggests that the relationship prompt guides the plain encoder to perform pixel-level classifications by encoding semantics in the `[CLS]` token, which should not be disregarded when obtaining segmentation results.

Table 7: **Ablation study on three different training-free projection modules (unit: %).**

| $\text{TFPM}_a$ | $\text{TFPM}_b$ | $\text{TFPM}_c$ | pAcc | $\text{mIoU}_s$ | $\text{mIoU}_u$ | hIoU |
|---|---|---|---|---|---|---|
| ✓ | | | 86.8 | 83.5 | 58.7 | 68.9 |
| | ✓ | | 94.5 | 92.4 | 77.6 | 84.4 |
| | | ✓ | 95.1 | 92.4 | 81.7 | 86.7 |

## 5 Conclusion

In this work, we propose the **R**elationship **P**rompt **M**odule (**RPM**) to guide VLM to transform its image-level embeddings into pixel-level semantic ones. RPM and VLM combine to form **R**elationship **P**rompt **N**etwork (**RPN**), a VLM-based OVSS method that directly performs OVSS without any explicit segmentation-specific networks. To the best of our knowledge, we are the first to give a straightforward solution for OVSS that applies prompt learning solely to VLM. We evaluate our method on four public benchmark datasets in both zero-shot and open-vocabulary settings, and achieve the state-of-the-art performance with only about **3M** trainable parameters (2% of total parameters). Therefore, it is concluded that VLM with relationship prompt learning is enough for open-vocabulary semantic segmentation without any explicit segmentation-specific networks.

**Limitations.** Although we meticulously design an effective prompt learning method for directly using VLM to achieve pixel-level OVSS, there are several ways for prompt learning to achieve further improvement: 1) directly acting on the attention map (more direct); 2) dynamically reorganizing multi-head attention map (more lightweight).

## Acknowledgements.

This work is supported by the National Natural Science Foundation of China (No. 62176224, 62176092, 62222602, 62306165, 62106075, 62476090, 62376233, 62431004); Natural Science Foundation of Shanghai (23ZR1420400); Natural Science Foundation of Chongqing (NO.CSTB2023NSCQ-JQX0007); Natural Science Foundation of Fujian Province under Grant 2024J09001; China Computer Federation (CCF) Lenovo Blue Ocean Research Fund; China Academy of Railway Sciences No.2023YJ357; Xiaomi Young Talents Program.

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

# Appendix

## A   Our Structure *vs* Existing Structures

We show the comparison of our method and the existing VLM-based methods in Figure 8. In our method, we abandon the explicit segmentation-specific networks, i.e., the mask proposal network and the semantic decoder and guide VLM to directly output the segmentation results. In existing VLM-based methods, the entire OVSS framework consists of VLM, the mask proposal network and the semantic decoder, incurring extensive training cost. As illustrated in Section 1, the two-stage methods usually adopt knowledge distillation to transfer zero-shot learning ability to the segmentation-specific network, and the one-stage methods utilize feature adaptation to train the semantic decoder. In summary, prompt learning acts as the engine to drive VLM to directly achieve OVSS in our method. Conversely, in the existing VLM-based methods, VLM acts as the engine to drive the segmentation-specific network to indirectly achieve OVSS. Therefore, our method is more straightforward and more parameter-efficient.

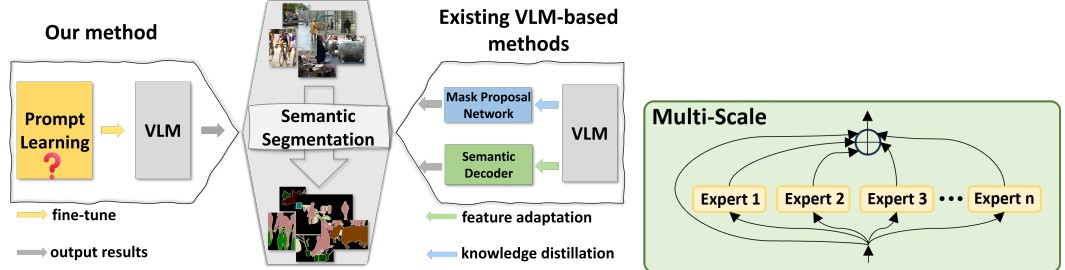

Figure 8: **Our method *vs* existing VLM-based methods.**       Figure 9: **Multi-Scale.**

## B   M2oE

**Details**   The role of the gating network is to select the right expert for each sample in a batch. Thus, we need to calculate the gating scores $G(x) \in \mathbb{R}^{B \times n}$, i.e., the scores for $B$ samples in a batch to $n$ experts. For simplicity, let the input, i.e., the patch embeddings $\mathbf{p}$, be denoted by $x \in \mathbb{R}^{B \times N \times d}$, where $N = h \times w$. The input $x$ is first directed into a common linear to reduce the dimension $d$ to $r$, and then is reshaped to the size of $\mathbb{R}^{B \times r}$ after global average pooling (as Eq. 12). Like in [42, 43], the value $H(x) \in \mathbb{R}^{B \times n}$ is calculated as follows:

$$x_a = \text{Reshape}(\text{AvgPool}(\underset{\mathbf{d}\rightarrow\mathbf{r}}{\mathbf{Linear}}(x)), (B, r)) \tag{12}$$

$$H(x) = (x_a \cdot W_g) + \text{StandardNormal}() \cdot \text{Softplus}(x_a \cdot W_{noise}) \tag{13}$$

where $W_g \in \mathbb{R}^{r \times n}$ and $W_{noise} \in \mathbb{R}^{r \times n}$ represent the trainable gating weight and the noise term, respectively. We select only the top $k$ values on $H(x)$ and set the rest to $-\infty$. After 'Softmax' operation, the gating scores $G(x)$ can be calculated as follows:

$$G(x) = \text{Softmax}(\text{KeepTopK}(H(x), k)) \tag{14}$$

The role of the experts is to extract feature with multi-scale, which consist of Interpolate operation at a specific scale, $\text{DWConv}_{3 \times 3}$ and Upsample operation to map the feature to the original scale. Let $i$-th expert be $E_i$ with scale $s_i = \frac{1}{2^{i-1}}$, where $i = 1, 2, 3, 4$. The expert $E_i$ processes the input $x$ as follows:

$$E_i(x) = \text{UpSample}(\text{DWConv}_{3 \times 3}(\text{Interpolate}(\underset{\mathbf{d}\rightarrow\mathbf{r}}{\mathbf{Linear}}(x), s_i))) \tag{15}$$

Finally, we obtain the output according to Eq. 1.

**Multi-Scale**   An intuitive alternative is applying a multi-scale strategy, which utilizes $n$ branches to extract features with different scales. The structural difference with M2oE is its lack of the gating network, as illustrated in Figure 9. The absence brings a higher parameter consumption, due to the fact that the gating network helps selectively activate sparse experts.

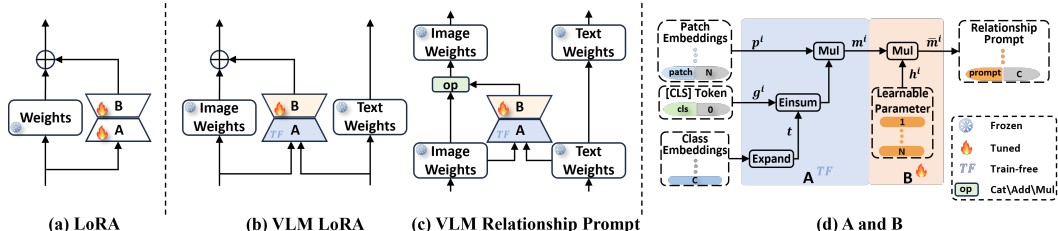

**(a) LoRA**      **(b) VLM LoRA**      **(c) VLM Relationship Prompt**      **(d) A and B**

Figure 10: **Relationship prompt.** (a) LoRA. (b) VLM LoRA in a common-bypass mode. (c) VLM relationship prompt tuning. (d) Details of **A** and **B**. Note that **A** and **B** refer to ours IPT and APG.

## C   Motivation behind the design of RPM

The motivation behind the design of RPM is inspired by LoRA [58]. As illustrated in Figure 10 (a), considering that the main components of large models lie in a low intrinsic dimension, LoRA introduces dimensionality reduction matrix **A** and dimensionality enhancement matrix **B** into the bypass to fit the frozen weights. This approach enables fine-tuning the model with minimal training costs, thereby improving performance for specific tasks without changing the original parameters. Given that VLM has two branches to process image and text data, we initially adopt a common-bypass mode as shown in Figure 10 (b). The distinction between LoRA and VLM LoRA lies in the fact that matrix **A** in VLM LoRA accepts two types of modal inputs and it is training-free. It is important to note that we do not connect the output of matrix **B** with the embeddings from the text module for two reasons: their dimensions do not match and the relationship attention map from matrix **A** performs more effectively as a visual prompt rather than a textual prompt, as illustrated in Figure 1. However, the performance of implementing VLM LoRA was suboptimal. We discovered that a direct skip connection with the output of the current image weights that bypasses matrices **A** and **B**, achieves the same effect. Our analysis suggests that merely tuning the lightweight matrix **B** in a post-tuning model is found to be insufficient for adapting parameter-intensive weights. Consequently, we propose a prefix-tuning architecture as illustrated in Figure 10 (c). We position the two matrices between the preceding and current weights and established a skip connection with the original output of the preceding image weights rather than the current image weights. This circumvents the need to fit the parameter-intensive image weights. In light of this prefix-tuning architecture, we categorize this model into the category of prompt learning rather than adaptation, and thereby naming it relationship prompt learning. In addition, we consider three operations between the output of the preceding image weights and the output of matrix **B**: concatenation, addition, and Hadamard product. Following a quantitative analysis (shown in Table 14), we select the concatenation operation.

## D   More Details about Experiments

**Zero-Shot Semantic Segmentation Setting**    We adopt the popular zero-shot setting as follows in[13, 9, 4, 3, 1, 31]. In the setting, we divide all classes into seen and unseen, only the seen classes are used in training. For the VOC dataset, we select 15 seen classes and 5 unseen classes, with the 'background' class excluded. For the COCO dataset, we divide all classes into 156 seen classes and 15 unseen classes. For the Context dataset, we select 50 seen classes (including 'background') and 10 unseen classes. These unseen classes are defined as follows:

| Dataset | Unseen Classes |
|---------|----------------|
| VOC | pottedplant, sheep, sofa, train, tvmonitor |
| COCO | cow, giraffe, suitcase, frisbee, skateboard carrot, scissors, cardboard, clouds, grass playingfield, river, road, tree, wall concrete |
| Context | cow, motorbike, sofa, cat, boat, fence bird, tv monitor, keyboard, aeroplane |

**Open-Vocabulary Semantic Segmentation Setting**    We employ the open-vocabulary setting as follows in[14, 12, 51, 53]. In this setting, we train our model on the COCO dataset and evaluate it on other datasets, including the VOC dataset with 20 classes (PAS-20), the Context dataset with 59 and 459 classes (PC-59 and PC-459), the ADE20K dataset with 150 and 847 classes (A-150 and A-847).

**Text Prompt Templates**   For the VOC dataset, we apply a single template 'A photo of a {}'. For the Context and COCO dataset, we apply multiple templates as follows:

| |
|---|
| 'A photo of a {}.' ; 'A photo of a small {}.' ; 'A photo of a medium {}.' ; 'A photo of a large {}.' |
| 'This is a photo of a {}.' ; 'This is a photo of a small {}.' ; 'This is a photo of a medium {}.' ; 'This is a photo of a large {}.' |
| 'A {} in the scene.' ; 'A photo of a {} in the scene.' |
| 'There is a {} in the scene.' ; 'There is the {} in the scene.' |
| 'This is a {} in the scene.' ; 'This is the {} in the scene.' ; 'This is one {} in the scene.' |

**Full Experiment Results**   We show the full performance comparison of existing methods in the zero-shot and open-vocabulary semantic segmentation settings in Table 8 and Table 9.

**Comparison with PEFT Methods**   We explore some parameter-efficient fine-tuning (PEFT) methods with baseline (i.e., the CLIP with LPM): 1) fine-tuning entire baseline; 2) only fine-tuning LPM; 3) BitFit [70], a sparse-fine-tuning method where only the bias-terms of the model (or a subset of them) are modified; 4) Adapter [71], which inserts a trainable adapter module between the transformer layers. 5) VPT [24], which inserts trainable tokens to the input feature of each transformer layer; 6) LST [72], which trains a ladder-side network, a small and separate network that takes intermediate activations as input via shortcut connections (called ladders) from the backbone networks and makes predictions. 7) SSF [73], which trains scale and shift parameters to modulate the visual features. 8) LoRA [58], which inserts a trainable dimensionality reduction matrix and a dimensionality enhancement matrix in parallel to the frozen weights. As illustrated in Table 10, our method shows significant improvements compared to PEFT methods. This is attributed to the fact that these PEFT methods only focus on how to train the baseline with fewer parameters, suitable only for image-level feature representation tasks, while our method not only trains the baseline with fewer parameters, but also focuses on applying capabilities suitable for image-level feature representation tasks directly to pixel-level semantic segmentation tasks. In summary, existing PEFT methods mainly focus on fine-tuning the task-specific model to improve performance on that task, while our method enables VLM to directly perform semantic segmentation.

**Combination with VPT**   We explore the combination of our method and VPT in Table 11. When using VPT alone and in combination with our method, the number of trainable tokens in each layer is 400 and 40 respectively. Considering the impact of parameter initialization in VPT, the performance change after the combination is negligible.

Table 8: **Full performance comparison in the zero-shot setting (unit:%).** Here, the best results are shown in bold and the second-best results are underlined.

| Methods | VOC | | | | COCO | | | | Context | | | |
|---|---|---|---|---|---|---|---|---|---|---|---|---|
| | pAcc | mIoU$_s$ | mIoU$_u$ | hIoU | pAcc | mIoU$_s$ | mIoU$_u$ | hIoU | pAcc | mIoU$_s$ | mIoU$_u$ | hIoU |
| w/o self-training | | | | | | | | | | | | |
| SPNet [33] | - | 78.0 | 15.6 | 26.1 | - | 35.2 | 8.7 | 14.0 | - | - | - | - |
| ZS3Net [31] | - | 77.3 | 17.7 | 28.7 | - | 34.7 | 9.5 | 15.0 | 52.8 | 20.8 | 12.7 | 15.8 |
| CaGNet [30] | 80.7 | 78.4 | 26.6 | 39.7 | 56.6 | 33.5 | 12.2 | 18.2 | - | 24.1 | 18.5 | 21.2 |
| SIGN [28] | - | 75.4 | 28.9 | 41.7 | - | 32.3 | 15.5 | 20.9 | - | - | - | - |
| Joint [29] | - | 77.7 | 32.5 | 45.9 | - | - | - | - | - | 33.0 | 14.9 | 20.5 |
| ZegFormer [11] | - | 86.4 | 63.6 | 73.3 | - | 36.6 | 33.2 | 34.8 | - | - | - | - |
| ZegFormer+MAFT [51] | - | 91.5 | 80.7 | 85.7 | - | 36.4 | 40.1 | 38.1 | - | - | - | - |
| ZSSeg [12] | 90.0 | 83.5 | 72.5 | 77.5 | 60.3 | 39.3 | 36.3 | 37.8 | - | - | - | - |
| ZSSeg +MAFT [51] | - | 87.1 | 76.1 | 81.2 | - | 36.1 | 35.9 | 36.0 | - | - | - | - |
| ZegCLIP [19] | 94.6 | 91.9 | 77.8 | 84.3 | 62.0 | 40.2 | 41.4 | 40.8 | 76.2 | 46.0 | 54.6 | 49.9 |
| FreeSeg [10] | - | 91.9 | 78.6 | 84.7 | - | **42.4** | 42.2 | **42.3** | - | - | - | - |
| **RPN(ours)** | **95.8** | **93.1** | **84.6** | **88.6** | **64.4** | 40.8 | **42.8** | 41.8 | **76.4** | **47.7** | **58.7** | **52.6** |
| w/ self-training | | | | | | | | | | | | |
| SPNet [33] | - | 77.8 | 25.8 | 38.8 | - | 34.6 | 26.9 | 30.3 | - | - | - | - |
| ZS5Net [31] | - | 78.0 | 21.2 | 33.3 | - | 34.9 | 10.6 | 16.2 | 49.5 | 27.0 | 20.7 | 23.4 |
| CaGNet [30] | 81.6 | 78.6 | 30.3 | 43.7 | 56.8 | 35.6 | 13.4 | 19.5 | - | - | - | - |
| STRICT [34] | - | 82.7 | 35.6 | 49.8 | - | 35.3 | 30.3 | 34.8 | - | - | - | - |
| DiffMask [32] | - | 71.4 | 65.0 | 68.1 | - | - | - | - | - | - | - | - |
| ZSSeg [12] | 88.7 | 79.2 | 78.1 | 79.3 | 63.8 | 39.6 | 43.6 | 41.5 | - | - | - | - |
| ZegCLIP [19] | 95.1 | 91.8 | 82.2 | 86.7 | 68.8 | 40.6 | 54.8 | 46.6 | 77.2 | 46.6 | 65.4 | 54.4 |
| MaskCLIP† [8] | - | 88.8 | 86.1 | 87.4 | - | 38.1 | 54.7 | 45.0 | - | 44.4 | 66.7 | 53.3 |
| ZegCLIP† [19] | 96.2 | 92.3 | 89.9 | 91.1 | 69.2 | **40.7** | 59.9 | 48.5 | 77.3 | 46.8 | 68.5 | 55.6 |
| **RPN†(ours)** | **97.1** | **93.1** | **93.6** | **93.3** | **69.3** | 40.6 | **61.2** | **48.8** | **78.3** | **48.1** | **70.8** | **57.3** |
| fully supervised | | | | | | | | | | | | |
| ZegCLIP [19] | 96.3 | 92.4 | 90.9 | 91.6 | 69.9 | 40.7 | 63.2 | 49.6 | 77.5 | 46.5 | 78.7 | 56.9 |
| **RPN(ours)** | **97.2** | **94.0** | **94.6** | **94.3** | **70.8** | **41.1** | **64.1** | **50.5** | **78.7** | **48.5** | **80.1** | **60.4** |

Table 9: **Full performance comparison in the open-vocabulary setting (unit:%).** Here, the best results are shown in bold and the second-best results are underlined.

| Methods | VLM | Training Set | A-847 | PC-459 | A-150 | PC-59 | PAS-20 |
|---|---|---|---|---|---|---|---|
| SPNet [33] | - | VOC | - | - | - | 24.3 | 18.3 |
| ZS3Net[31] | - | VOC | - | - | - | 19.4 | 38.3 |
| LSeg[59] | ViT-B/32 | VOC-15 | - | - | - | - | 47.4 |
| LSeg+[60] | ALIGN | COCO-Stuff | 2.5 | 5.2 | 13.0 | 36.0 | - |
| Han et al.[61] | ViT-B/16 | COCO Panoptic [62] | 3.5 | 7.1 | 18.8 | 45.2 | 83.2 |
| GroupViT[63] | ViT-S/16 | GCC[64]+YFCC[65] | 4.3 | 4.9 | 10.6 | 25.9 | 50.7 |
| ZegFormer[11] | ViT-B/16 | COCO-Stuff | 5.6 | 10.4 | 18.0 | 45.5 | 89.5 |
| OpenSeg [60] | ALIGN | COCO Panoptic+Loc. Narr.[66] | 4.4 | 7.9 | 17.5 | 40.1 | - |
| FreeSeg [10] | - | COCO-Stuff | 7.1 | 6.4 | 17.9 | 34.4 | 85.6 |
| FreeSeg+MAFT [51] | - | COCO-Stuff | 10.1 | 12.8 | 29.1 | 53.5 | 90.0 |
| OVSeg [14] | ViT-B/16 | COCO-Stuff+COCO Caption | 7.1 | 11.0 | 24.8 | 53.3 | 92.6 |
| CAT-Seg[52] | ViT-B/16 | COCO-Stuff | 8.4 | 16.6 | 27.2 | **57.5** | 93.7 |
| SAN[53] | ViT-B/16 | COCO-Stuff | 10.1 | 12.6 | 27.5 | 53.8 | 94.0 |
| SED[54] | ConvNeXt-B | COCO-Stuff | 11.4 | **18.6** | **31.6** | 57.3 | 94.4 |
| **RPN(ours)** | ViT-B/16 | COCO-Stuff | 11.4 | 17.3 | 31.5 | 57.1 | **95.2** |
| LSeg[59] | ViT-B/32 | VOC-15 | - | - | - | - | 52.3 |
| OpenSeg [60] | ALIGN | COCO Panoptic+Loc. Narr. | 8.1 | 11.5 | 26.4 | 44.8 | - |
| OVSeg [14] | ViT-L/14 | COCO-Stuff+COCO Caption | 9.0 | 12.4 | 29.6 | 55.7 | 94.5 |
| Ding et al.[67] | ViT-L/14 | COCO Panoptic | 8.2 | 10.0 | 23.7 | 45.9 | - |
| ODISE[68] | ViT-L/14 | COCO Panoptic | 11.1 | 14.5 | 29.9 | 57.3 | - |
| HIPIE[69] | BERT-B | COCO Panoptic | - | - | 29.0 | 59.3 | - |
| CAT-Seg[52] | ViT-L/14 | COCO-Stuff | 10.8 | 20.4 | 31.5 | **62.0** | 96.6 |
| SAN[53] | ViT-L/14 | COCO-Stuff | 13.7 | 17.1 | 33.3 | 60.2 | 95.5 |
| FC-CLIP[55] | ConvNeXt-L | COCO Panoptic | 14.8 | 18.2 | 34.1 | 58.4 | 95.4 |
| SED[54] | ConvNeXt-L | COCO-Stuff | 13.9 | **22.6** | 35.2 | 60.6 | 96.1 |
| **RPN(ours)** | ViT-L/14 | COCO-Stuff | **14.9** | 22.1 | **36.4** | 61.9 | 96.6 |

Table 10: **Performance comparison with PEFT methods (unit:%).** Baseline represents the CLIP model with LPM. **#Params(M)** represents the number of trainable parameters during training.

| Methods | #Params(M) | VOC | | | | COCO | | | |
|---|---|---|---|---|---|---|---|---|---|
| | | pAcc | mIoU$_s$ | mIoU$_u$ | hIoU | pAcc | mIoU$_s$ | mIoU$_u$ | hIoU |
| Baseline | 154.5 | 84.1 | 83.5 | 31.2 | 45.4 | 47.8 | 30.1 | 19.6 | 23.7 |
| LPM-only | 0.8 | 88.8 | 87.3 | 45.3 | 59.6 | 49.3 | 31.3 | 20.3 | 24.6 |
| BitFit[70] | 4.0 | 89.7 | 79.3 | 51.2 | 62.2 | 50.6 | 35.6 | 23.3 | 28.2 |
| Adapter[71] | 3.9 | 90.3 | 79.7 | 51.6 | 62.6 | 51.4 | 36.0 | 24.1 | 28.9 |
| VPT[24] | 4.0 | 90.9 | 81.0 | 52.9 | 64.0 | 51.9 | 37.5 | 25.9 | 30.6 |
| LST[72] | 11.5 | 88.6 | 78.7 | 50.4 | 61.4 | 50.1 | 34.8 | 22.6 | 27.4 |
| SSF[73] | 4.4 | 90.8 | 80.8 | 52.7 | 63.8 | 51.7 | 37.3 | 25.6 | 30.4 |
| LoRA[58] | 4.0 | 91.3 | 82.2 | 53.1 | 64.5 | 52.9 | 38.7 | 27.0 | 31.8 |
| **RPN(ours)** | 3.2 | 95.8 | 93.1 | 84.6 | 88.6 | 64.4 | 40.8 | 42.8 | 41.8 |

Table 11: **Performance of combining our method and VPT (unit:%).** **#Params(M)** represents the number of trainable parameters during training.

| Methods | #Params(M) | VOC | | | | COCO | | | |
|---|---|---|---|---|---|---|---|---|---|
| | | pAcc | mIoU$_s$ | mIoU$_u$ | hIoU | pAcc | mIoU$_s$ | mIoU$_u$ | hIoU |
| VPT[24] | 4.0 | 90.9 | 81.0 | 52.9 | 64.0 | 51.9 | 37.5 | 25.9 | 30.6 |
| **RPN(ours)** | 3.2 | 95.8 | 93.1 | 84.6 | 88.6 | 64.4 | 40.8 | 42.8 | 41.8 |
| RPN+VPT | 3.6 | 95.8 | 93.1 | 84.2 | 88.4 | 64.4 | 40.3 | 43.1 | 41.7 |

Table 12: **Impact of different pre-trained weights for the plain encoder (unit:%).**

| Weights | VOC | | | | COCO | | | |
|---|---|---|---|---|---|---|---|---|
| | pAcc | mIoU$_s$ | mIoU$_u$ | hIoU | pAcc | mIoU$_s$ | mIoU$_u$ | hIoU |
| ViT[56] | 85.9 | 81.0 | 42.9 | 56.1 | 46.7 | 29.3 | 19.9 | 23.7 |
| MAE[74] | 87.1 | 82.4 | 43.7 | 57.1 | 48.2 | 30.8 | 21.5 | 25.3 |
| CLIP[5] | 95.8 | 93.1 | 84.6 | 88.6 | 64.4 | 40.8 | 42.8 | 41.8 |

Table 13: **Impact of RPM in different layers (unit:%).** 'Number' represent the number of layer at which RPM is applied. Note that 'all' do not include the last layer.

| Datasets | Number | pAcc | mIoU$_s$ | mIoU$_u$ | hIoU |
|---|---|---|---|---|---|
| VOC | 1 | 93.1 | 88.4 | 74.1 | 80.6 |
| | 11 | 92.8 | 88.7 | 70.3 | 78.4 |
| | {1,3,5,7,9,11} | 93.9 | 89.4 | 80.9 | 84.9 |
| | {2,4,6,8,10} | 93.9 | 89.1 | 80.1 | 84.4 |
| | all | 95.8 | 93.1 | 84.6 | 88.6 |
| COCO | 1 | 60.2 | 39.8 | 35.8 | 37.7 |
| | 11 | 60.1 | 39.4 | 33.6 | 36.3 |
| | {1,3,5,7,9,11} | 61.9 | 40.2 | 38.5 | 39.3 |
| | {2,4,6,8,10} | 61.9 | 40.1 | 38.4 | 39.2 |
| | all | 64.4 | 40.8 | 42.8 | 41.8 |

Table 14: **Ablation study on three kinds of prefix-tuning operations (unit:%).**

| Datasets | Mul | Add | Cat | pAcc | mIoU$_s$ | mIoU$_u$ | hIoU |
|---|---|---|---|---|---|---|---|
| VOC | ✓ | | | 88.8 | 87.3 | 45.3 | 59.6 |
| | | ✓ | | 90.3 | 83.2 | 69.5 | 75.7 |
| | | | ✓ | 95.8 | 93.1 | 84.6 | 88.6 |
| COCO | ✓ | | | 49.3 | 31.3 | 20.3 | 24.6 |
| | | ✓ | | 56.8 | 36.6 | 25.7 | 30.2 |
| | | | ✓ | 64.4 | 40.8 | 42.8 | 41.8 |

**Impact of Different Pre-trained Weights for the Plain Encoder**    We explore different pre-trained weights of the plain encoder in Table 12. These weights are from supervised learning, unsupervised learning and weakly-supervised learning of text signals. ViT [56] applies supervised learning to learn visual representation. MAE [74] applies self-supervised learning to learn rich semantics. The two methods learn uni-modal knowledge. CLIP[5] applies text as a supervisory signal and adopts contrastive learning to learn general visual representation. It is a multi-modal learning method. The results show that our method has limitations in mining uni-modal visual knowledge. In addition, compared to ViT and MAE, CLIP uses more data to pre-train the plain encoder. This also shows that even though our method is superior, it still cannot get rid of the dependence on a large amount of data.

**Impact of RPM in Different Layers**    We adopt a layer-by-layer guidance mode (i.e., all) as shown in Table 13. Our method is evaluated across various network depths, including shallow layers, deep layers, and interval layer mode. We find that the performance of single-layer mode (such as layers 1 or 11) are significantly worse than those of multi-layer modes. The layer-by-layer mode demonstrates optimal performance on both the VOC and COCO datasets.

**Impact of Different Prefix-tuning Operations**    We compare three prefix-tuning operations as shown in Figure 10 (c): concatenation, addition, and Hadamard product. Results from Table 14 indicate that the relationship prompt should not directly modify the original feature (like addition or Hadamard product), but rather should influence the model indirectly by computing the attention between the prompt and the original feature (like concatenation).

# E    Visualization

Baseline represents the frozen CLIP model with LPM.

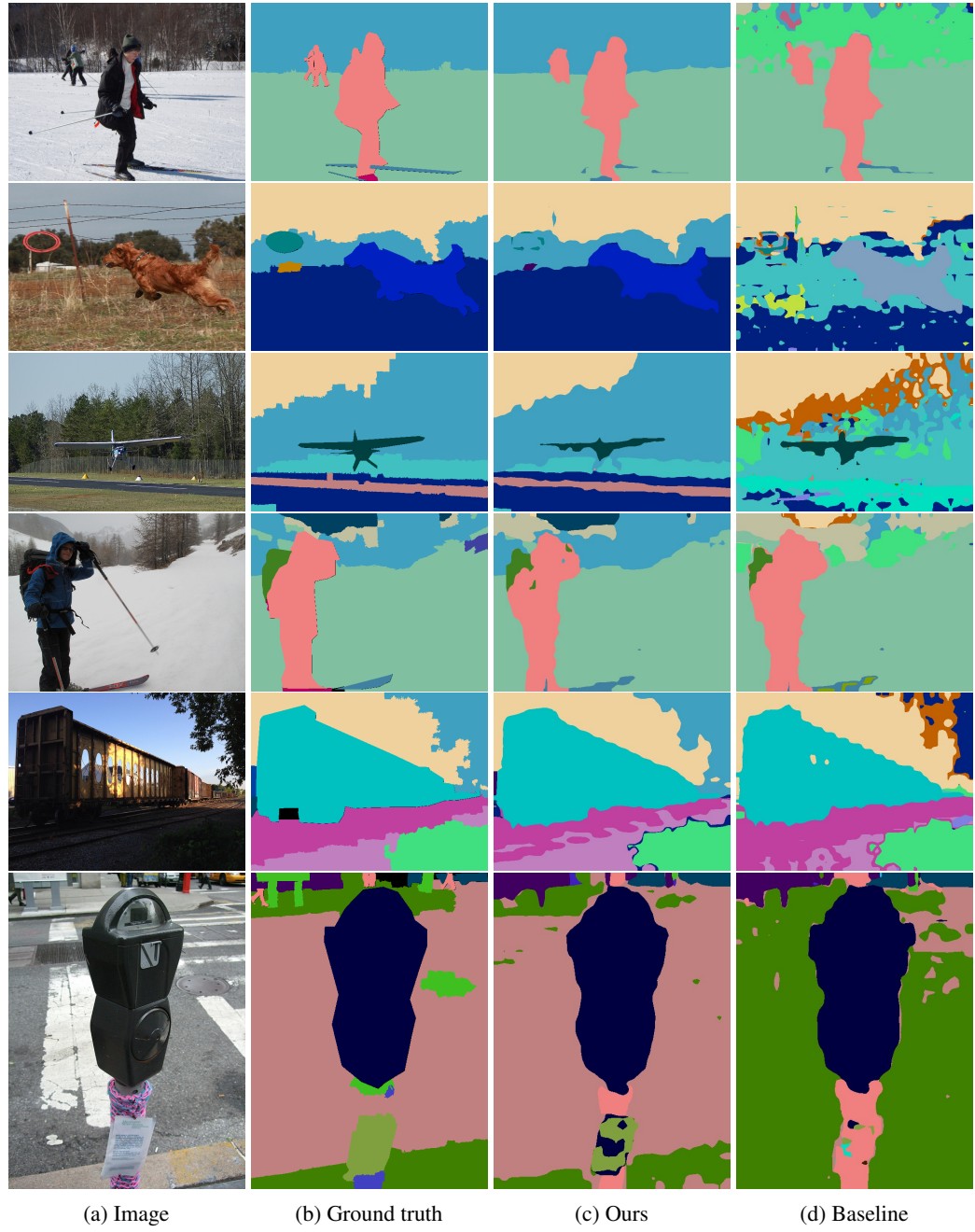

(a) Image      (b) Ground truth      (c) Ours      (d) Baseline

Figure 11: **Qualitative analysis.** The unseen classes include tree , frisbee , grass and road .

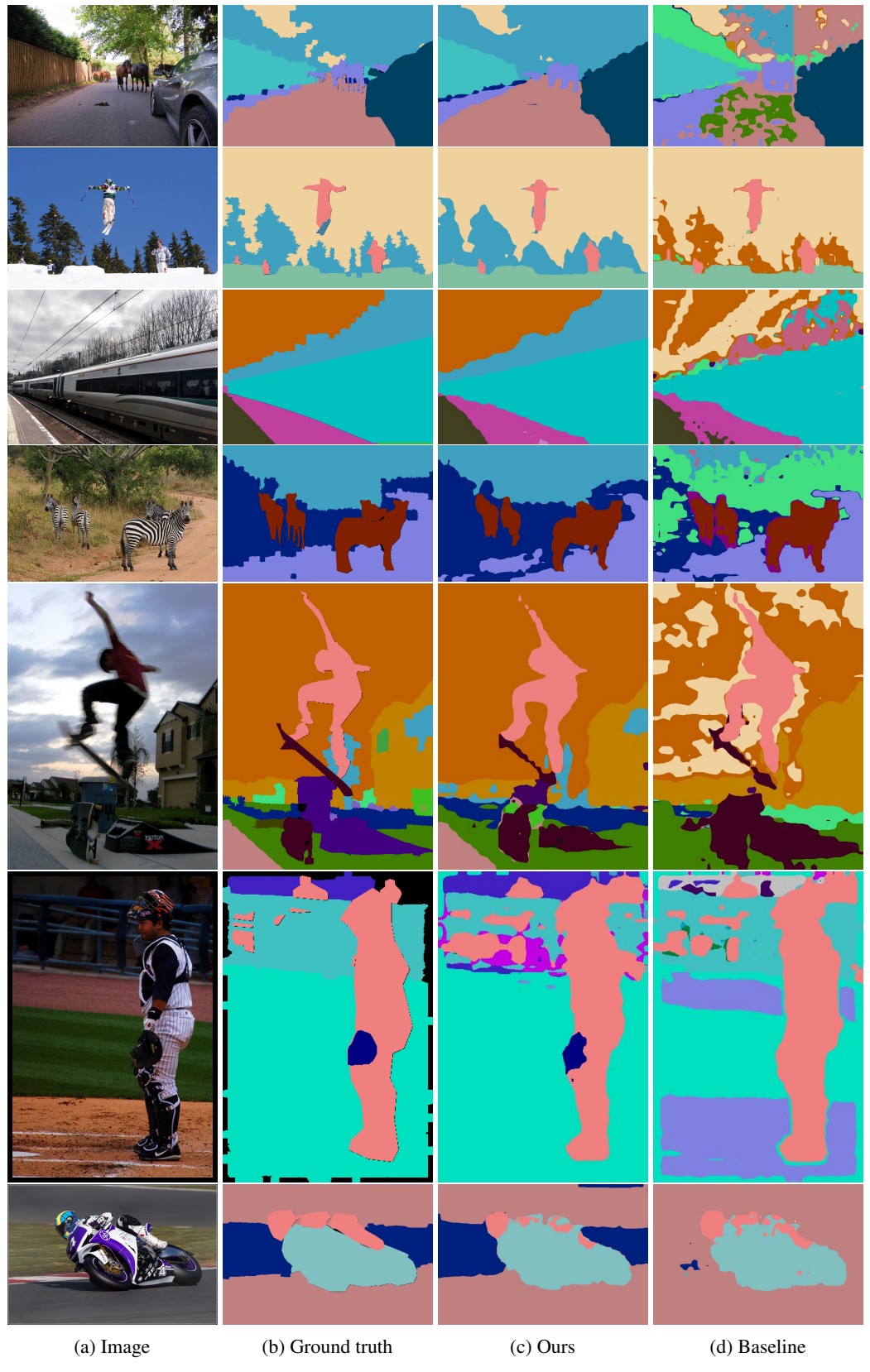

|               |                  |          |              |
|:-------------:|:----------------:|:--------:|:------------:|
| (a) Image     | (b) Ground truth | (c) Ours | (d) Baseline |

Figure 12: **Qualitative analysis.** The unseen classes include `tree`, `clouds`, `skateboard`, `playingfield`, `wall-concrete`, `grass` and `road`.

