# OpenReview forum: "Relationship Prompt Learning is Enough for Open-Vocabulary Semantic Segmentation"
_NeurIPS.cc/2024/Conference — NeurIPS 2024 poster_

### Official Review · Reviewer_Rvjz · 2024-07-09

**Soundness:** 2
**Presentation:** 1
**Contribution:** 2
**Rating:** 4
**Confidence:** 4

**Summary:**

This paper tackles open-vocabulary semantic segmentation, a task aiming at per-pixel predictions of image inputs on any given classes. Existing methods are built on pre-trained vision-language models like CLIP. Different from CLIP that achieves coarse open-vocabulary classification, open-vocabulary segmentation extends this understanding capability to pixel level. This paper claims that existing approaches take much compute for training. Authors develop a Relationship Prompt Network that features dense image-text correlations across all layers. Also, their approach includes some training techniques, including mixture-of-experts, multi-scale input and prompt learning. Experimental results show that proposed RPN achieves performance gains against published methods like ZegCLIP [19], MaskCLIP [8] and FreeSeg [10].

**Strengths:**

- **Performance.** The proposed RPN outperforms several published methods, such as ZegCLIP, MaskCLIP and FreeSeg.
- **Inference efficiency.** Table 1 shows that proposed RPN is trained with fewer learnable parameters. In inference, RPN shows less computation in FLOPs.

**Weaknesses:**

- **Unclear unique contributions.** I am not quite understand about the unique contribution from this paper. Does this paper give new insights to the community? The proposed relationship prompt is a combination of powerful tuning methods that have been proven effective, including dense image-text attentions, mixture-of-experts, multi-scale inputs and prompt learning. None of these modules are emphasized and I cannot see the unique value of applying these techniques. Performance gains from these techniques does not surprise me that much.
- **Many typos and confusing claims**. Please refer to Questions 1-11 for details. I think these issues should be taken care of before considered for publication.

**Questions:**

- Line 32, “it is challenging to apply prompt learning solely to VLM for OVSS”. Has this claim been checked? Why applying prompt learning to VLM for OVSS is challenging? Does “Challenging” refer to “technically challenging”, right?
- Line 49-51, “we find they can construct an image-text relationship attention map via the attention mechanism”. Image-text relationship attention map is a confusing claim. This should refer to a well defined phrase in this field, such as language-aware deep fusion in [A], which is an influential paper in this field.
- Figure 1, how are these attention maps plotted?
- Line 58-61, we propose linear projection module comprising merely two individual linear layers, which maps the image and text feature into a shared space, and then computes their matrix product to produce the results. What should matrix product refer to? Element-wise or column-wise?
- Typos. Line 67 and 69, RPN to RPM. There is potential confusion between RPM and RPN.
- Line 113-114, reducing parameter consumption. I do not think this is a correct claim. Adding dense cross-attentions will surely increase parameters for inference. I do not see techniques in this paper that reduces parameters like quantization.
- Line 113-125, refer to a figure here is better for illustration purpose.
- Figure 2, what is M2oE, ITP and APG block? A bit confusing.
- Table 4, what does the first line mean and what is the difference between the first and the second line?
- Figure 7, only legends of head 1 and 2 are given, while others are missing. In such case, I cannot understand what is presented in this figure.
- Table 1, instead of params, this should be Learnable params, right?

[A] Grounded Language-Image Pre-training. CVPR 2022.

**Limitations:**

Authors have included limitations of proposed RPN in Appendix D.

---

> ### Author Rebuttal · Authors · 2024-08-07
>
> # Response to Reviewer Rvjz
> Thank you so much for acknowledging the strength of our method. We have carefully considered your constructive and insightful comments and here are the answers to your concerns.
>
> **Q1. Unclear unique contributions.**
> Please refer to **General Response-Q1**.
>
> **Q2. Confusing claim that *it is challenging to apply prompt learning solely to VLM for OVSS* should be argued.**
> We would like to re-clarify the claim as:
> We point out two reasons for the claim (in L.32-41): (1) Current prompt provides only image-level granularity, which restricts VLMs from performing tasks related to pixel-level visual localization. (2) A lack of an image-text relationship, which hinders the exploration of VLMs’ potential for open-vocabulary scene understanding. To demonstrate this, we compare seven current PEFT methods applied to the baseline (in L.618-638), and show the comparison in Tab.9 and 10. We show the part of Tab.9 as:
> |(VOC)|#Params(M)|pAcc|mIoU$_s$|mIoU$_u$|hIoU|
> |:-|:-:|:-:|:-:|:-:|:-:|
> |Baseline|154.5|84.1|83.5|31.2|45.4|
> |VPT|4.0|90.9|81.0|52.9|64.0|
> |LoRA|4.0|91.3|82.2|53.1|64.5|
> |**RPN(ours)**|**3.2**|**95.8**|**93.1**|**84.6**|**88.6**|
>
> We would like to re-emphasize the conclusion as: it is challenging to apply prompt learning solely to VLM for OVSS, and our method shows significant improvement (at least 20% mIoU improvement in VOC).
>
> **Q3. Concerns between image-text relationship attention map and language-aware deep fusion in [1\*].**
> We analyzed their differences from three perspectives as follows:
> (1) Motivation: We aim to directly output pixel-level predictions using only the frozen VLM, while [1*] aim to proposal a grounded language-image pre-training method.
> (2) Structure: We use a train-free method to construct our map, while [1*] uses a trainable cross-attention module to construct it; we only need to introduce our map for the image encoder, while [1*] also introduces it for the text encoder. Thus, our method is more concise and direct.
> (3) Training cost: we do not need additional training sets, parameter-intensive adapter and large-scale pre-training, which are all required by [1*].
> In summary, our method achieves OVSS with lower training cost and simpler structure.
>
> [1*] Grounded Language-Image Pre-training.
>
> **Q4. Fig.1, how are these attention maps plotted?**
> We directly visualize the results of eq.2, ${{m}}^i = {p}^i \cdot  (t \odot {g}^i)^\top$, using a heat map.
>
> **Q5. What should matrix product refer to? Element-wise or column-wise?**
> Let the image and text feature denote $p \in \mathbb{R}^{n\times d}$ and $t\in \mathbb{R}^{c\times d}$. The matrix product $O\in \mathbb{R}^{n\times c}$ between them $O_{ij} = \sum_{k=1}^d p_{ik}t^\top_{kj}$.
>
> **Q6. Typos. Line 67 and 69, RPN to RPM. There is potential confusion between RPM and RPN.**
> In fact, RPM and RPN refers to **R**elationship **P**rompt **M**odule and **R**elationship **P**rompt **N**etwork. As illustrated in L.61-62, RPN consists of RPM and VLM.
>
> **Q7. Line 113-114, concerns about reducing parameter consumption.**
> We clarify the sentence of L.113-114 that our method only add 3M trainable parameters with the frozen VLM to achieve OVSS, so our method is low-cost during training. As illustrated in Tab.1, we conduct efficiency experiment to demonstrate our method indeed consume very small amount of additional trainable parameters to achieve OVSS. In addition, we add the efficiency for ADE20K and Context as:
> ||#Params(M)|FLOPs(G)|FPS|
> |:-|:-:|:-:|:-:|
> |ADE20K|3.2|117.5|10.4|
> |Context|3.2|95.5|10.7|
>
> **Q8. L.113-125, refer to a figure here is better for illustration purpose.**
> Thanks for your suggestion.
>
> **Q9. Fig.2, what is M2oE, ITP and APG block? A bit confusing.**
> Actually, Fig.2 has nothing to do with these blocks. Are you referring to Fig.3? If yes, we have mentioned these blocks in the legend of Fig.3. RPM consists of M2oE, ITP and APG block. M2oE refers to multi-scale mixture-of-experts, which aims to propose multi-scale vision knowledge (L.134-150). ITP refers to image-to-pixel semantic attention, which aims to enable the image encoder to learn open-vocabulary semantics from text features (L.151-168). APG refers to adaptive prompt generation, which aims to construct the adaptive image-text relationship for each pixel (L.169-179).
>
> **Q10. Tab.4, what does the first line mean and what is the difference between the first and the second line?**
> RPM (the first line) denotes the combination of M2oE, ITP and APG. Therefore, RPM without M2oE (the second line) denotes the ablation about APG and ITP.
>
> **Q11. Concerns about Fig.7. Only legends of head 1 and 2 are given, while others are missing.**
> First, we give all heads in Fig.7 (MAD evaluation) instead of head 1 and 2, and use different color point to denote different heads; due to the number of heads is large, only the symbols for the two heads are given in the legend, and the remaining heads are indicated by ellipses (note the legend at the bottom right).
> Second, MAD can be used to explore the range of attention of each attention head, similar to the receptive field in convolutional neural networks (CNNs) (e.g., Fig.7 in ViT[1*] is also represented to valuate MAD). It is a common metric. A higher point indicates a larger receptive field, and greater point spacing signifies richer feature diversity. Fig.7 show that with the guidance of relationship prompt, the deep features of VLM gradually have a wider MAD value range, which indicates fine-grained semantic properties.
>
> [1*] An Image is Worth 16x16 Words: Transformers for Image Recognition at Scale.
>
> **Q12. Tab.1, instead of params, this should be Learnable params, right?**
> Yes, params denotes learnable params.

---

> > ### Comment · Reviewer_Rvjz · 2024-08-09
> >
> > Thanks authors for detailed responses to my concerns. Most typo concerns have been addressed, but surely need better clarifications and clearer presentation in the future manuscript. I do have a few previous concerns regarding contributions and novelties for this paper, which I think is more important and have not addressed by this rebuttal, and also a few new ones which need better clarifications.
> >
> > ### Reply to Q1 (General Response-Q1)
> >
> > 1.  I am not quite sure if I clearly understand “additional training sets”. Why are these training sets “additional”?
> >
> > 2.  Another thing is, performing parameter-efficient segmentation on top of a frozen VLM (or fine-grained understanding) is very straightforward [A,B]. I do not think this is a novel motivation.
> >
> > ### Reply to Q2
> >
> > 1. Authors claim that "current prompt … a lack of an image-text relationship, which hinders the exploration of VLM potential for open-vocabulary scene understanding." This is not an accurate claim, considering paper [C].
> >
> > In all, I still do not see the unique value from this paper. Many positive factors in this paper that could lead to performance gains, as I included in **Weaknesses 1**. I will adjust my score based on responses from other reviewers and how authors clarify their unique contributions.
> >
> > [A] F-VLM: Open-Vocabulary Object Detection upon Frozen Vision and Language Models. ICLR’23.
> >
> > [B] Convolutions Die Hard: Open-Vocabulary Segmentation with Single Frozen Convolutional CLIP. NeurIPS’23.
> >
> > [C] MaPLe: Multi-modal Prompt Learning. CVPR’23.

---

> > > ### Author Response · Authors · 2024-08-10
> > >
> > > Thanks for your exhaustive reply and suggestions. Here we have three more clarifications:
> > >
> > > **Q1: I am not quite sure if I clearly understand “additional training sets”. Why are these training sets “additional”?**
> > > There are two main ways to achieve OVSS using VLM: (1) Training VLM with pixel-level supervision from scratch allows it to directly output pixel-level predictions, which means that the process needs to introduce “additional” pixel-level supervised training sets, like SegGPT[1*] use a mixture of ADE20K, COCO, PASCAL VOC, Cityscapes and so on to train ViT to directly output segmentation results. (2) Unlike training from scratch, adopting knowledge distillation or feature adaptation to transfer the frozen VLM's semantic knowledge to segmentation-specific networks dose not introduce “additional” pixel-level supervised training sets, except COCO under the OVSS setting requirements. Thus, the former usually needs “additional training sets; and the latter needs additional segmentation-specific networks.
> > >
> > > [1*] SegGPT: Towards Segmenting Everything In Context
> > >
> > > **Q2: Performing parameter-efficient segmentation on top of a frozen VLM (or fine-grained understanding) is very straightforward. I do not think this is a novel motivation.**
> > > As answered in **General Response-Q1**, training from scratch relies on additional training sets and large-scale pre-training, while only needs a simple VLM; adopting knowledge distillation or feature adaptation relies on additional segmentation-specific networks based on VLM, while only needs small-scale fine-tuning. Thus, is it possible to combine the advantages of both, using small-scale fine-tuning while only requiring a simple VLM (discarding the complex segmentation-specific networks)? This is exactly what we do. We would like to re-emphasize our contribution as: **we only use VLM with extremely low training cost (almost 3M trainable parameters) to directly achieve OVSS. (Please note that we do not rely on segmentation-specific networks and only need parameter-efficient fine-tuning)**. We would like to use the table in **General Response-Q1** again to illustrate the difference between our approach and the methods you listed.
> > > ||Additional Training Sets|Parameter-intensive Decoder Adapter|Large-scale Pixel-level Pre-training|
> > > |:-|:-:|:-:|:-:|
> > > |F-VLM|❌|✔|❌|
> > > |FC-CLIP|❌|✔|❌|
> > > |RPN(ours)|❌|❌|❌|
> > >
> > > **Q3: Authors claim that "current prompt … a lack of an image-text relationship, which hinders the exploration of VLM potential for open-vocabulary scene understanding." This is not an accurate claim, considering paper[1\*].**
> > > As illustrated in L.107-108, MaPLe adopts trainable prompt to guide both visual and textual embeddings and proposes a coupling function as a bridge to build a multi-modal prompt. The coupling function is implemented as a linear layer, which maps the text prompt to the image prompt. Note that the mapping process of text-to-image prompt completely relies on text prompt without introducing the image embeddings of the VLM. In addition, the generation of text prompt also relies solely on a randomly initialized learnable vector that is independent of the text embeddings of VLM. Therefore, in essence, MaPLe does not construct a prompt with image-text relationship, but only generates a text prompt from a randomly initialized learnable vector and maps it to an image prompt with the help of a linear layer.
> > >
> > > [1*] MaPLe: Multi-modal Prompt Learning.

---

> > > > ### Comment · Reviewer_Rvjz · 2024-08-10
> > > >
> > > > Thanks authors for further detailed and thorough clarifications. My confusion remains and I think they may need further discussions.
> > > >
> > > > For **Q1**, I firstly get confused because you include GroupViT in **Global Response Q1** to a category of approaches that use "additional" training sets. You use SegGPT as an example to explain this, which appears very confusing to me, because SegGPT and GroupViT do not even fall into the same category (GroupViT does not apply pixel-level supervision while SegGPT does). For training multimodal segmentation network, which dataset do you perceive as "additional", "image-caption pairs" or "pixel-wise masks"?
> > > >
> > > > For **Q2**, one suggestion for authors is to include number of parameter (#param) for F-VLM, FC-CLIP and proposed RPN.
> > > >
> > > > For **Q3**, authors claim that "MaPLe does not construct a prompt with image-text relationship, but only generates a text prompt from a randomly initialized learnable vector and maps it to an image prompt with the help of a linear layer." I think "good image-text relationship" is a very vague claim and does not convince me that much, lacking theoretical or empirical justifications. Many CLIP-related papers study how to build "good or better image-text relationship". Why RPN is better than them? Also, why authors think "generates text prompt from randomly initialized learnable vectors" is a worse try, any justification?

---

> > > > > ### Author Response · Authors · 2024-08-11
> > > > >
> > > > > Thank you for your prompt reply. We would like to further discuss.
> > > > >
> > > > > **Response for Q1**
> > > > > Compared to training a pre-trained model, training a vanilla model from scratch means introducing a large-scale training set (either pixel-level supervision or image-level supervision), so this large-scale training set is "additional" to the training process of the pre-trained model. For GroupViT, it use CC and YFCC as "additional" image-level supervised training sets; for SegGPT, it use a mixture of ADE20K, COCO, PASCAL VOC, Cityscapes and so on as "additional" pixel-level supervised training sets.
> > > > >
> > > > > **Response for Q2**
> > > > > According to your comments, the comparison with FC-CLIP and F-VLM in #params is shown as (note that F-VLM does not provide the corresponding number of trainable parameters, so we roughly estimate the value based on their open source code for reference only):
> > > > > |(ADE20K)|#Params(M)|
> > > > > |:-|:-:|
> > > > > |FC-CLIP|21|
> > > > > |F-VLM|38|
> > > > > |RPN(ours)|3.2|
> > > > >
> > > > > **Response for Q3**
> > > > > We would like to further elaborate on our point of view.
> > > > > First, we would like to illustrate the relational modeling capabilities of RPN to answer the question “Why RPN is better than them?”. The relational capability of the model is reflected in guiding the model to make pixel-level predictions for unseen classes by establishing relationship attention between text class information and pixel-level visual information, making its performance close to that of seen classes. We visualize the relationship attention map in Fig.1. The first line denote the relationship attention maps for seen class across each layer; the last two lines for unseen classes. For seen class, the model has prior pixel-level semantic information, so the relationship attention map only needs to focus on a small number of pixels to guide the model to make class predictions for these pixels (e.g., the relationship attention map for airplane has fewer highlighted areas). For unseen class, the model lacks corresponding pixel-level semantic information, so the relationship attention map needs to focus on more pixels to provide the model with more sufficient pixel-level semantic information (e.g., the shallow relationship attention map for sheep highlights the complete semantic information). In addition, to demonstrate the guidance mechanism, we conduct related interpretability experiments (i.e., MAD evaluation), as illustrated in L.271-282. The MAD results show that with the guidance of relationship prompt, the deep features of VLM gradually have a wider MAD value range, which indicates fine-grained semantic properties. We also conduct system level comparison as shown in Tab.2,3, 7 and 8 to demonstrate the superior performance.
> > > > > Second, generating prompt from randomly initialized learnable vectors, e.g., VPT[1*], make poor performance (shown in Tab.10). We think that the randomly initialized learnable prompt cannot be associated with the VLM’s prior image-text knowledge and therefore cannot effectively guide the VLM to perform pixel-level prediction directly.
> > > > >
> > > > > [1*] Visual prompt tuning.

---

> > > > > > ### Author Response · Authors · 2024-08-12
> > > > > >
> > > > > > Dear Reviewer,
> > > > > >
> > > > > > Thank you again for reviewing our manuscript. We have tried our best to address your questions, and revised our paper by following suggestions from all reviewers.
> > > > > >
> > > > > > Please kindly let us know if you have any follow-up questions or areas needing further clarification. Your insights are valuable to us, and we stand ready to provide any additional information that could be helpful.

---

> ### Comment · Reviewer_Rvjz · 2024-08-12
>
> Thanks for authors prompt and active replies. For **Q2**, I think presented results convince me a bit. However, in spite of performance gains presented in this paper and many turns of debating, I am still confused on unique contributions from this paper and remain doubts on its novelty. Authors seem to claim very high-level/vague contributions (could claimed by most multi-modal learning papers) without theoretical or empirical justifications and unique insights, which do not convince me that much. So far I am inclined to reject this paper and waiting other reviewers response for a second opinion.

---

> > ### Author Response · Authors · 2024-08-12
> >
> > Thanks for your patient response and comments. We would like to reiterate our contribution: we only use image-level supervised VLM (note that no segmentation-specific network is introduced) and achieve OVSS with extremely low training cost. We are the first to do so. Although we did not reach a consensus in the end, thank you for your discussion and wish you all the best in your future research.

---

### Official Review · Reviewer_Rcdd · 2024-07-12

**Soundness:** 4
**Presentation:** 3
**Contribution:** 3
**Rating:** 7
**Confidence:** 4

**Summary:**

The paper proposes a training and inference-efficient Relationship Prompt Network (RPN). This network leverages a layer-wise Relationship Prompt Module (RPM) utilizing tuning methods similar to VLM LoRA and an improved Linear Projection Module (LPM) without relying on a segmentation model. The authors conduct extensive experiments, demonstrating the solid efficiency and effectiveness of their approach.

**Strengths:**

The paper proposes a method for continuously injecting pixel-level image-text relationships into the layers of the image encoder using efficient tuning techniques. The experiments conducted are very comprehensive.

**Weaknesses:**

The paper lacks a discussion on region-text relationships, focusing instead on directly leveraging pixel-text relationships.
Additionally, there should be a discussion and experiments with other visual prompt tuning methods applied to the baseline.

**Questions:**

See the weakness.

**Limitations:**

Please provide a more detailed analysis on the correlation between the relational abilities of the model and its performance in open-vocabulary tasks (separating seen and unseen classes).

---

> ### Author Rebuttal · Authors · 2024-08-07
>
> # Response to Reviewer Rcdd
> Thank you so much for acknowledging the strength of our method. We have carefully considered your constructive and insightful comments and here are the answers to your concerns.
>
> **Q1. Lack of a discussion on region-text relationships, focusing instead on directly leveraging pixel-text relationships.**
> Thanks for your valuable suggestion. The discussion on region-text relationship is worthy.
> First, we clarify the difference between pixel-text and region-text relationship: the former links image pixel with text description; the latter links image mask with text description. Since image mask contains more information, it can more intuitively represent things or stuff, while the image pixels are the opposite. Thus, image mask can be more easily associated with text description. For example, region-text relationship can be established by directly matching image mask and text description, while when we need to establish pixel-text relationship after featuring image mask and text description.
> Second, we consider the impact of region-text on OVSS. Although constructing region-text relationship seems to be simpler, is it more effective for OVSS? As illustrated in MaskFormer[1*], pixel-level classification is not necessarily better than mask-level classification for semantic segmentation, which seems to show that region-text relationship enable the VLM to achieve OVSS better than pixel-text relationship. However, adopting region-text relationship means that VLM need to output mask proposals, which are usually output by a DERT-like[1*,2*] decoder. This shows that VLM need a DERT-like decoder, and goes against our intended goal of achieving OVSS using only VLM without any segmentation-specific networks.
> Third, we consider the impact of region-text relationship on open-vocabulary segmentation. VLM with region-text relationship can output class-agnostic mask, thus making it feasible to achieve instance segmentation and object detection. This is a very interesting direction worth exploring, and it gives us some inspiration for future work.
>
> [1*] Per-Pixel Classification is Not All You Need for Semantic Segmentation.
> [2*] End-to-End Object Detection with Transformers.
>
> **Q2. There should be a discussion and experiments with other visual prompt tuning methods applied to the baseline.**
> Thanks for your valuable suggestion. In fact, we have discussed seven current PEFT methods (including other visual prompt tuning) applied to the baseline as illustrated in L.618-638, and show the comparison in Tab.9 and 10. We show the part of Tab.9 as:
> |(VOC)|#Params(M)|pAcc|mIoU$_s$|mIoU$_u$|hIoU|
> |:-|:-:|:-:|:-:|:-:|:-:|
> |Baseline|154.5|84.1|83.5|31.2|45.4|
> |VPT|4.0|90.9|81.0|52.9|64.0|
> |LoRA|4.0|91.3|82.2|53.1|64.5|
> |**RPN(ours)**|**3.2**|**95.8**|**93.1**|**84.6**|**88.6**|
>
> We would like to re-emphasize the conclusion as: our method shows significant improvement (at least 20% mIoU improvement in VOC), due to that existing PEFT methods mainly focus on fine-tuning the image-supervised VLM to improve performance on classification. But, our method enables VLM to directly achieve OVSS.
>
> **Q3. Please provide a more detailed analysis on the correlation between the relational abilities of the model and its performance in open-vocabulary tasks (separating seen and unseen classes).**
> Thanks for your valuable suggestion. We provide our analysis as follows:
> The relational capability of the model is reflected in guiding the model to make pixel-level predictions for unseen classes by establishing relationship attention between text class information and pixel-level visual information, making its performance close to that of seen classes. We visualize the relationship attention map in Fig.1. The first line denote the relationship attention maps for seen class across each layer; the last two lines for unseen classes. For seen class, the model has prior pixel-level semantic information, so the relationship attention map only needs to focus on a small number of pixels to guide the model to make class predictions for these pixels (e.g., the relationship attention map for airplane has fewer highlighted areas). For unseen class, the model lacks corresponding pixel-level semantic information, so the relationship attention map needs to focus on more pixels to provide the model with more sufficient pixel-level semantic information (e.g., the shallow relationship attention map for sheep highlights the complete semantic information). In addition, to demonstrate the guidance mechanism, we conduct related interpretability experiments (i.e., MAD evaluation), as illustrated in L.271-282. The MAD results show that with the guidance of relationship prompt, the deep features of VLM gradually have a wider MAD value range, which indicates fine-grained semantic properties. We also conduct system level comparison (separating seen and unseen classes) as shown in Tab.2 and 7.

---

> > ### Author Response · Authors · 2024-08-12
> >
> > Dear Reviewer,
> >
> > Thank you again for reviewing our manuscript. We have tried our best to address your questions, and revised our paper by following suggestions from all reviewers.
> >
> > Please kindly let us know if you have any follow-up questions or areas needing further clarification. Your insights are valuable to us, and we stand ready to provide any additional information that could be helpful.

---

### Official Review · Reviewer_2wFX · 2024-07-13

**Soundness:** 2
**Presentation:** 3
**Contribution:** 2
**Rating:** 4
**Confidence:** 5

**Summary:**

This paper proposes relationship prompt module (RPM), which generates relationship prompt that directs VLM to extract pixel-level semantic embeddings suitable for OVSS. Moreover, RPM integrates with VLM to construct relationship prompt network (RPN), achieving OVSS without segmentation-specific networks. RPN attains state-of-the-art performance with merely about 3M trainable parameters (2% of total parameters).

**Strengths:**

1.  This paper proposes RPM, which generates pixel-level relationship prompt to guide VLM to transform image-level embeddings to pixel-level ones suitable for OVSS.

2. This paper proposes a simple yet parameter-efficient OVSS method, i.e., RPN, employing relationship prompt learning solely to perform OVSS without any segmentation-specific networks.

3. RPN attains state-of-the-art results on four public benchmarks by optimizing about 3M trainable parameters (2% of total parameters).

**Weaknesses:**

1. In the experiment tables, there is no Efficiency comparison for ADE20K and Context dataset.

2. The ablation studies in Table 4 are insufficient; they only include M2oE and LPM ablation studies, and do not cover ITP and APG.

3. The performance is not state-of-the-art compared to some previous works, such as CAT-Seg: Cost Aggregation for Open-Vocabulary Semantic Segmentation.

**Questions:**

Please refer to the concerns and issues raised in the "Weaknesses".

**Limitations:**

Please refer to the concerns and issues raised in the "Weaknesses".

---

> ### Author Rebuttal · Authors · 2024-08-07
>
> # Response to Reviewer 2wFX
> Thank you so much for acknowledging the strength of our method. We have carefully considered your constructive and insightful comments and here are the answers to your concerns.
>
> **Q1. No efficiency comparison for ADE20K and Context dataset.**
> We have verified the efficiency on two benchmarks as illustrated in L.245-249. According to your comments, we add the efficiency for ADE20K and Context as:
> |     | #Params(M) | FLOPs(G) |	FPS |
> | :------- | :------------: | :-------: |:-------: |
> |ADE20K|3.2|117.5|10.4|
> |Context|3.2|95.5|10.7|
>
> **Q2. The ablation study in Tab.4 do not cover ITP and APG.**
> We would like to re-clarify the ablation study in Tab.4. Tab.4 mainly shows the impact of M2oE, ITP, APG and LPM (we use RPM to denote the combination of M2oE, ITP and APG). Although we do not directly mark APG and ITP in Tab.4, RPM without M2oE denotes the ablation about APG and ITP. Therefore, we have conducted the component of APG and ITP. In addition, due to APG and ITP together form our relationship prompt (ablating one of them will cause bug), it is notable that APG and ITP can not be separated for ablation.
>
> **Q3. The performance is not SOTA compared to some previous works.**
> Thanks for your suggestions, please refer to **General Response-Q2**.

---

> > ### Author Response · Authors · 2024-08-12
> >
> > Dear Reviewer,
> >
> > Thank you again for reviewing our manuscript. We have tried our best to address your questions, and revised our paper by following suggestions from all reviewers.
> >
> > Please kindly let us know if you have any follow-up questions or areas needing further clarification. Your insights are valuable to us, and we stand ready to provide any additional information that could be helpful.

---

### Official Review · Reviewer_DSMk · 2024-07-15

**Soundness:** 2
**Presentation:** 2
**Contribution:** 2
**Rating:** 4
**Confidence:** 5

**Summary:**

This paper primarily studies Open-Vocabulary Semantic Segmentation. The main contribution of this paper is the proposal of RPN, which employs relationship prompt learning solely to perform OVSS without any segmentation-specific networks. The authors claim that RPN attains state-of-the-art results on four public benchmarks.

**Strengths:**

The strengths of this paper can be listed as follows,

- The paper clearly expresses its main research content and the proposed algorithm.

- The proposed algorithm achieved good results on multiple benchmark datasets.

**Weaknesses:**

The weaknesses of this paper can be summarized as follows,

- The authors claim that their method achieved state-of-the-art results, but the results in Table 3 and Table 8 do not support this claim. Additionally, the methods listed in Tables 2, 3, 7 and 8 are somewhat limited. More recent works should be included to substantiate the claim that the proposed method achieves state-of-the-art results. (refer to https://paperswithcode.com/task/open-vocabulary-semantic-segmentation)

- The authors should provide a more detailed explanation of the content in Figure 2, either in the caption or in the introduction section. (Or, they could reference this figure in a more suitable location, such as the first paragraph of the introduction section. Or, the authors could mention that a more detailed explanation is provided in the supplementary materials.)

- According to the authors' definition, there have already been many works that achieve open vocabulary semantic segmentation without segmentation-specific networks, e.g., "Image Segmentation Using Text and Image Prompts", "GroupViT: Semantic Segmentation Emerges from Text Supervision", "CAT-Seg: Cost Aggregation for Open-Vocabulary Semantic Segmentation", "SegGPT: Towards Segmenting Everything In Context", to name a few. These works, to some extent, diminish the novelty of the authors' contribution. (To be clear, I am not saying that the authors' work lacks novelty entirely, but its novelty is not as strong as claimed in the paper.)

- Did the authors conduct any ablation studies on the hyper-parameters that appear in Equations 9 to 11?

- Why do the authors use class embeddings from only text layer 12, while using image embeddings from various image layers? Can the authors provide some analysis or experimental support for this choice?

- When visualizing the attention map in Figure 1, the authors could also include a color legend to represent the response intensity.

- Figures 4 to 6 should include annotations to explain the meanings of the various symbols used in the images.

- Did the authors conduct any ablation studies on how to select the scale and the number of experts used in M2oE?

- Is the Hadamard product in Equation 2 significant? Would using a matrix product between $p^i$​ and $g^i$ instead affect the results?

- Why is pixel-level information used instead of patch-level information?

- Does the proposed algorithm produce block artifacts when generating segmentation maps for high-resolution images?

Overall, the paper is quite dull and does not stand out in the field of open vocabulary semantic segmentation. However, for a conference paper, the amount of work presented seems sufficient.

**Questions:**

I have placed all the questions I want to ask in the weaknesses box. Overall, the paper doesn't seem to have any major issues, but personally, I find the work uninteresting and believe it offers limited insights to the field of open vocabulary semantic segmentation. However, if other reviewers feel that this work is worth accepting, I wouldn't strongly oppose it.

**Limitations:**

The authors did not discuss the limitations of their proposed algorithm in either the main text or the supplementary materials.
I have highlighted some limitations of the paper in the weaknesses box. The authors can refer to these points to further improve the quality of the paper.

---

> ### Author Rebuttal · Authors · 2024-08-07
>
> # Response to Reviewer DSMk
> Thank you so much for acknowledging the strength of our method. We have carefully considered your constructive and insightful comments and here are the answers to your concerns.
>
> **Q1. The claim that our methods achieved SOTA results lack persuasiveness.**
> Thanks for your suggestions, please refer to **General Response-Q2**.
>
> **Q2. More detailed explanation of the content in Fig.2.**
> Thanks for your suggestion. We will modify the description in L.63 as: *Fig.2 shows the key difference between RPN and other OVSS methods. RPN makes VLM directly output pixel-level predictions by prompt learning, while other methods use VLM to assist a complex segmentation-specific networks where VLM transfers its rich knowledge to the mask proposal network by knowledge distillation or make the semantic decoder output segmentation masks by feature adaptation.*
>
> **Q3. Some works that achieve OVSS without segmentation-specific networks diminish the novelty of our contribution.**
> Thanks for your suggestions, please refer to **General Response-Q1**.
>
> **Q4. Ablation studies about the hyper-parameters $\lambda_1, \lambda_2$ in eq.11.**
> $\lambda_1, \lambda_2$ control the weights between $\mathcal{L}\_{focal}$ and $\mathcal{L}_{dice}$. We explore three common combinations. The results is shown as:
> |$\lambda_1$|$\lambda_2$|A-847|PC-459|A-150|PC-59|PAS-20|
> |:-|:-:|:-:|:-:|:-:|:-:|:-:|
> |20|1|11.0|16.9|31.1|56.7|95.1|
> |1|1|10.4|16.7|29.6|55.4|94.8|
> |**100**|**1**|**11.4**|**17.3**|**31.5**|**57.1**|**95.2**|
>
> The results show that $\lambda_1=100$ and $\lambda_2=1$ are the best.
>
> **Q5. Why using class embeddings from only text layer 12, not others?**
> To explore which text layer of class embeddings to use, we conduct studies as:
> ||pAcc|mIoU$_s$|mIoU$_u$|hIoU|
> |:-|:-:|:-:|:-:|:-:|
> |one-to-one|69.7|65.4|17.3|27.4|
> |only last|95.8|93.1|84.6|88.6|
>
> One-to-one denotes using the same-layer embeddings from image and text encoder, and only last denotes using the last class embeddings, i.e., layer 12. We find that the shallow text embeddings cannot effectively guide VLM to achieve OVSS. The reason is that the shallow embeddings cannot accurately represent the class information, thereby constructing wrong relationship prompt.
>
> **Q6. A color legend is needed in Fig.1**
> Thanks for your suggestion. We will add the color legend in the future version. The degree of attention from low to high is marked by colors from dark blue to red. The more attention, the darker red; the less attention, the darker blue.
>
> **Q7. Explanation about some symbols in Fig. 4 to 6 is needed.**
> We will add these explanation in caption as: *$\odot$, $\otimes$ and $\oplus$ denotes element-wise product, matrix product and addition. The symbols Expand, Einsum and Mul denote expanding class dimension, element-wise product and matrix product.*
>
> **Q8. Ablation studies about the scale and the number of experts used in M2oE.**
> Note that the scale and number of experts are same parameter, which control the size of feature maps. Due to we crop images into $512\times512$, the scale can be set to 1/8 of the input feature maps at most. Thus, the scale $s_i = \frac{1}{2^{i-1}}$, where $i \in [1,4]$, i.e., the maximum number of experts is 4. The ablation studies is shown as:
> |$i$ |pAcc|mIoU$_s$|mIoU$_u$|hIoU|
> |:-|:--:|:-:|:-:|:-:|
> |(1,2,3,4)|95.8|93.1|84.6|88.6|
> |(1,2,3)|95.6|92.9|83.8|88.1|
> |(1,2)|95.6|92.8|83.5|87.9|
>
> The results show that embeddings with more diverse scales can improve OVSS performance.
>
> **Q9. Is the Hadamard product in eq.2 significant? Would using a matrix product between $p^i$​ and $g^i$ instead affect the results?**
> We rewrite eq.2 as: ${{m}}^i = {p}^i \cdot  (t \odot {g}^i)^\top$, where $p^i \in \mathbb{R}^{n\times d}$, $t\in \mathbb{R}^{c\times d}$ and $g^i \in \mathbb{R}^{1\times d}$. Please note the calculation process figure in our top-level pdf for more details.
> Hadamard product have an important effect in eq.2. Eq.2 (including Hadamard product and matrix product) can construct pixel-text relationship attention map to guide image encoder to transform image-level semantic to pixel-level. Firstly, Hadamard product assigns class weights to images in a batch by fusing $t$ used to identify different classes and $g^i$ used to identify each image in a batch, thus attaining image-text relationship. Based on this relationship, matrix product gets the class distribution of a pixel, which is normalized to one. Because $p^i$ contains pixel-level visual information, eq.2 achieves pixel-text relationship. Fig.1 visualizes the guidance process from image-text to pixel-text relationship. To demonstrate the image-to-pixel guiding scheme, we also conduct related interpretability experiments (i.e., MAD evaluation, L.271-282). The MAD results show that with the guidance of relationship prompt, the deep features of VLM gradually have a wider MAD value range, which indicates fine-grained semantic properties.
> Due to that we need relationship map $m^i \in \mathbb{R}^{n\times c}$, thus matrix product between $p^i$​ and $g^i$ cannot output map of this shape. In fact, we have explored matrix product between $p^i$​ and $t$ in L.291-311 and Tab.5. The results show that missing any of the product operations will have a significant impact on performance.
>
> **Q10. Why is pixel-level information used instead of patch-level information?**
> In fact, pixel-level information for feature maps used refers to patch-level information for original images. Due to that ViT usually divides the image into multiple patches for encoding, a pixel of the feature describes a patch of the image.
>
> **Q11. Does the proposed algorithm produce block artifacts when generating segmentation maps for high-resolution images?**
> No, for high-resolution images, we adopt *slide inference* mode, i.e., set stride to $341\times 341$, and crop size to $512\times 512$.

---

> > ### Author Response · Authors · 2024-08-12
> >
> > Dear Reviewer,
> >
> > Thank you again for reviewing our manuscript. We have tried our best to address your questions, and revised our paper by following suggestions from all reviewers.
> >
> > Please kindly let us know if you have any follow-up questions or areas needing further clarification. Your insights are valuable to us, and we stand ready to provide any additional information that could be helpful.

---

### Author Rebuttal · Authors · 2024-08-07

# General Response

We would like to thank all reviewers for providing constructive feedback that helped us improved the paper. We are encouraged that reviews think our paper:
* Clear expression: *The paper clearly expresses its main research content and the proposed algorithm.* (Reviewer DSMk)
* Good performance: *The proposed algorithm achieved good results on multiple benchmark datasets.* (Reviewer DSMk) *RPN attains state-of-the-art results on four public benchmarks.* (Reviewer 2wFX) *The proposed RPN outperforms several published methods.* (Reviewer Rvjz) *RPN shows less computation in FLOPs.* (Reviewer Rvjz)
* Comprehensive experiments: *The experiments conducted are very comprehensive.* (Reviewer Rcdd)

We have been working diligently on improving the paper on several fronts, addressing your critique. Below, we summarize the changes that we have made in an updated draft.
* More explanation about the diagrams. (Please refer to Response to Reviewer DSMk-Q2,6-7)
* More discussion on region-text relationship. (Please refer to Response to Reviewer Rcdd-Q1)
* More analysis on the correlation between the relational abilities of the model and its performance. (Please refer to Response to Reviewer Rcdd-Q3)

We respond to each of the reviewers in detail, and to some common issues below.

**(Reviewer DSMk and Rvjz) Q1. Concerns about contributions.**
We would like to re-emphasize our novelty as follows:
(1) Motivation: There are two main ways to achieve OVSS using image-level supervised VLM: pre-training the VLM from scratch using pixel-level supervision; using knowledge distillation or feature adaptation to assist in training parameter-intensive segmentation-specific networks with VLM. However, the training cost of these methods is high (the former needs additional training sets and large-scale pixel-level pre-training; the latter needs additional parameter-intensive segmentation-specific networks). We aim to froze VLM and just make prompt learning to achieve OVSS. Thus, the training is low-cost.
(2) Difference: Although there exist OVSS methods which do not rely on segmentation-specific networks, they do not just rely on frozen VLM to achieve OVSS. For example, CLIPSeg[1*], SED[3*] and CAT-Seg[2*] all rely on an additional parameter-intensive decoder adapter; GroupViT[4*] and SegGPT[5*] both need additional segmentation training sets. We would like to re-emphasize our contribution as: **we only use VLM with extremely low training cost (almost 3M trainable parameters) to directly achieve OVSS.** The comparison of important aspects such as, training sets, decoder adapter and pixel-level pre-training, between RPN and the other OVSS methods without segmentation-specific networks are shown as:
||Additional Training Sets|Parameter-intensive Decoder Adapter|Large-scale Pixel-level Pre-training|
|:-|:-:|:-:|:-:|
|CLIPSeg|❌|✔|❌|
|SED|❌|✔|❌|
|CAT-Seg|❌|✔|❌|
|GroupViT|✔|❌|✔|
|SegGPT|✔|❌|✔|
|RPN(ours)|❌|❌|❌|

[1*] Image Segmentation Using Text and Image Prompts
[2*] CAT-Seg: Cost Aggregation for Open-Vocabulary Semantic Segmentation
[3*] SED: A Simple Encoder-Decoder for Open-Vocabulary Semantic Segmentation
[4*] GroupViT: Semantic Segmentation Emerges from Text Supervision
[5*] SegGPT: Towards Segmenting Everything In Context

**(Reviewer DSMk and 2wFX) Q2. The claim that our methods achieved SOTA results lack persuasiveness.**
There are usually two types of experimental settings (zero-shot and open-vocabulary settings) for OVSS, involving multiple benchmarks. To fully evaluate the performance, we conducted experiments in the two settings.

Firstly, in zero-shot settings, our method outperforms existing methods on all benchmarks in Tab.2 and 7. Secondly, in open-vocabulary setting, our method outperforms CAT-Seg[2*] (the latest OVSS method) on three benchmarks (A-847, PC-459 and A-150), and achieve the highest results on three benchmarks (A-847, PAS-20 and A-150) in Tab.3 and 8. Thirdly, in both the two settings, we show the comparison as:
||A-849|pc-459|A-150|PC-59|PAS-20|VOC(mIoU$_u$)|COCO(mIoU$_u$)|Context(mIoU$_u$)|
|:-|:-:|:-:|:-:|:-:|:-:|:-:|:-:|:-:|
|SPNet|-|-|-|24.3|18.3|15.6|8.7|-|
|ZS3Net|-|-|-|19.4|38.3|17.7|9.5|12.7|
|ZegFormer|5.6|10.4|18.0|45.5|89.5|63.6|33.2|-|
|FreeSeg|7.1|6.4|17.9|34.4|85.6|78.6|42.2|-|
|RPN(ours)|**11.4**|**17.3**|**31.5**|**57.1**|**95.2**|**84.6**|**42.8**|**58.7**|

The result show that our method achieve SOTA on all benchmarks in both the two settings.

---

### Decision · Program_Chairs · 2024-09-25

**Decision:**

Accept (poster)

**Comment:**

Overall the reviews found the contribution in terms of technique too small, and the evaluation of benefit lacking for several reasons.  There was some confusion about acronyms that seems reasonable given the authors' choice of using coining two, RPM and RPN, that are similar to each other and to other acronyms used in the field.  The ability to adapt a VLM, without an additional explicit segmentation model, to come close to the state of the art is hopefully sufficient with some careful clarification in the paper text.